# Mettl3-mediated mRNA m⁶A modification controls postnatal liver development by modulating the transcription factor Hnf4a

Yan Xu [1,2,4], Zhuowei Zhou [1,4], Xinmei Kang[1,4], Lijie Pan[1,3], Chang Liu[1], Xiaoqi Liang[1,3], Jiajie Chu[1,3], Shuai Dong[1], Yanli Li[1], Qiuli Liu[1], Yuetong Sun[1], Shanshan Yu[1,3] & Qi Zhang [1,2,3] ✉

Hepatic specification and functional maturation are tightly controlled throughout development. N6-methyladenosine (m⁶A) is the most abundant RNA modification of eukaryotic mRNAs and is involved in various physiological and pathological processes. However, the function of m⁶A in liver development remains elusive. Here we dissect the role of Mettl3-mediated m⁶A modification in postnatal liver development and homeostasis. Knocking out Mettl3 perinatally with *Alb-Cre* (*Mettl3* cKO) induces apoptosis and steatosis of hepatocytes, results in severe liver injury, and finally leads to postnatal lethality within 7 weeks. m⁶A-RIP sequencing and RNA-sequencing reveal that mRNAs of a series of crucial liver-enriched transcription factors are modified by m⁶A, including *Hnf4a*, a master regulator for hepatic parenchymal formation. Deleting Mettl3 reduces m⁶A modification on *Hnf4a*, decreases its transcript stability in an Igf2bp1-dependent manner, and down-regulates *Hnf4a* expression, while overexpressing Hnf4a with AAV8 alleviates the liver injury and prolongs the lifespan of *Mettl3* cKO mice. However, knocking out Mettl3 in adults using *Alb-Cre^ERT2* does not affect liver homeostasis. Our study identifies a dynamic role of Mettl3-mediated RNA m⁶A modification in liver development.

The liver is the primary organ responsible for metabolism, lipid transportation, drug detoxification, and hormone secretion[1]. Hepatic specification and the dramatic functional transition from haematopoiesis to metabolism of the liver are tightly controlled by the intricate crosstalk among extracellular signals, transcription factors, and epigenetic regulators. Previous reports showed that coordination of liver-enriched transcription factors, histone modifications, and DNA methylation orchestrates the hepatic differentiation program during development and maintains liver homeostasis in adults[2,3].

N6-methyladenosine (m⁶A) modification has been identified as the most abundant modification in eukaryotic mRNAs since its discovery in the 1970s[4–6]. Dynamic deposition and removal of m⁶A is catalyzed by the methyltransferase complex containing methyltransferase-like 3 (METTL3), methyltransferase-like 14 (METTL14)[7,8], and Wilms tumor 1-associating protein (WTAP)[9], along with other co-factors and demethylases, including ALKB homolog 5 (ALKBH5)[10] and fat mass and obesity-associated protein (FTO)[11], respectively. m⁶A modification is recognized by YTH family members which embrace the domain of YT521-B homology (YTHDF1-3, YTHDC1-2)[12–16] and insulin-like growth factor-2 mRNA-binding proteins (IGF2BP1/2/3)[17], and are thus involved in various steps of RNA metabolism, such as stability, translation, nuclear exportation, splicing of mRNAs, and biogenesis and maturation of miRNAs[18].

[1]Biotherapy Centre, The Third Affiliated Hospital, Sun Yat-sen University, Guangzhou, China. [2]Guangdong Provincial Key Laboratory of Liver Disease Research, The Third Affiliated Hospital, Sun Yat-sen University, Guangzhou, China. [3]Cell-gene Therapy Translational Medicine Research Centre, The Third Affiliated Hospital, Sun Yat-sen University, Guangzhou, China. [4]These authors contributed equally: Yan Xu, Zhuowei Zhou, Xinmei Kang. ✉e-mail: zhangq27@mail.sysu.edu.cn

In this work, we generate hepatic-specific *Mettl3* knockout (*Mettl3* cKO) mice by crossing *Mettl3^{flox/flox}* mice with *Albumin* (*Alb*)-enhancer/promoter driven-*Cre* transgenic mice to investigate the role of m⁶A modification in liver development. Hepatic perinatal loss of Mettl3 causes severe liver damage, including steatosis, apoptosis, and fibrosis, and finally results in lethality within 7 weeks. Using m⁶A-RNA immunoprecipitation (m⁶A-RIP) sequencing and RNA-sequencing, we identify that crucial liver-enriched transcription factors, including *Hnf4a*, are modified by m⁶A in liver development. Loss of Mettl3 induces depletion of m⁶A on *Hnf4a* transcripts, decreases its transcript stability in an Igf2bp1-dependent manner, and down-regulates *Hnf4a* expression, while overexpressing Hnf4a with AAV8 alleviates the liver injury and prolongs lifespan of *Mettl3* cKO mice. However, deletion of Mettl3 in adult mouse livers using *Albumin*-enhancer/promoter-driven *Cre^{ERT2}* shows minimal effects on liver homeostasis. In conclusion, we elucidate a dynamic role of Mettl3-mediated RNA m⁶A modification during mouse postnatal liver development and decipher a novel function of epitranscriptomic control of liver organogenesis.

## Results

### Generation of hepatic specific Mettl3 knockout mice

To study the role of m⁶A modification in liver development, we first tested the expression level of critical subunits of the m⁶A methyltransferase complex, Mettl3 and Mettl14[7]. Both components showed shallow protein levels in mouse neonates (within one day after birth), increased gradually, and peaked at 2-3 weeks, and then decreased from 4 weeks onwards (Supplementary Fig. 1a). A similar trend was observed in human livers with high expression in children and a subsequent decline with age (Supplementary Fig. 1b). These results indicated that m⁶A is dynamically regulated in postnatal liver development. Global knockout of either *Mettl3* or *Mettl14* results in embryonic lethality caused by gastrulation defects[19–21]. Thus, to study the role of m⁶A modification, we generated mice with hepatic specific knockout of the catalytic subunit of the m⁶A methyltransferase complex, Mettl3, by crossing *Mettl3^{flox/flox}* mice (with loxP sites flanking exons 2 and 4) with *Alb*-enhancer/promoter-driven *Cre* transgenic mice (Supplementary Fig. 1c, d). The specific knockout of Mettl3 in the liver was confirmed by genomic PCR, quantitative real-time PCR (RT-qPCR), western blot, and immunochemistry (Fig. 1a-d and Supplementary Fig. 1e-j). Genomic PCR and RT-qPCR showed that efficient knockout of Mettl3 started from day 1 after birth (Fig. 1b and Supplementary Fig. 1k), along with *Cre* expression (Supplementary Fig. 1i). As expected, livers from *Mettl3* cKO mice showed a significant decrease in mRNA m⁶A levels compared to control mice (Supplementary Fig. 1l). In addition, we also observed that knocking out Mettl3 led to disruption of Mettl14 (Supplementary Fig. 1m), which is in accordance with previous reports[19].

### Hepatic Mettl3 knockout results in postnatal lethality

Hepatic *Mettl3* knockout mice were born at almost expected Mendelian frequencies (Fig. 1e), excluding the possibility of prenatal lethality. However, both male and female knockout mice were smaller in body size than their age and sex-matched wild-type (WT) control (Control, also hereafter in similar experiments) littermates (Fig. 1f). This difference appeared at 2 weeks after birth and gradually became more pronounced at 4 and 5 weeks (Fig. 1g and Supplementary Fig. 1n, o). Moreover, all the *Mettl3* cKO mice died within 7 weeks after birth, while heterozygous knockout individuals were fertile and survived for over 12 months without discernible defects in development (Fig. 1h), indicating that one allele of *Mettl3* was sufficient to maintain normal development and function of mouse livers. These results demonstrate that Mettl3 is critical for postnatal liver development, especially during the highly proliferative stages from 0 to 4 weeks after birth.

### Hepatic Mettl3 deletion causes liver injury

To delineate the exact role of Mettl3 in liver organogenesis, we dissected the livers of Control and *Mettl3* cKO mice at different time points after birth. Grossly, an obvious mottled appearance, which is an indication of lipid deposition, was observed in the livers of *Mettl3* cKO mice 2 weeks after birth (Fig. 2a). The livers of 4-week-old *Mettl3* cKO mice were yellow, smaller, and stiffer than the WT Control. These differences became more significant at 5 weeks (Fig. 2a). The liver weight of *Mettl3* cKO mice decreased since 3 weeks after birth (Fig. 2b), while the liver weight to body weight ratio was slightly increased (Supplementary Fig. 2a). Serum indicators of liver function showed that Mettl3 deficiency caused defects in metabolism, detoxification, protein synthesis, and secretion functions in the liver from 1-2 weeks after birth, indicating progressive liver damage (Fig. 2c-j and Supplementary Fig. 2b).

### Mettl3 deletion results in apoptosis, steatosis, fibrosis, and activation of hepatic progenitors

Although histologic analysis showed no apparent differences between the Control and *Mettl3* cKO mice at 1 day and 1 week after birth (Supplementary Fig. 3a, b), marked pathological lesions were observed in *Mettl3* cKO mice since 2 weeks old (Fig. 3a). Increased lipid droplet deposition in *Mettl3* cKO livers was observed 2 weeks after birth, confirmed by BODIPY staining and Oil Red O staining for both frozen liver tissues (Fig. 3b and Supplementary Fig. 3c, d) and primary hepatocytes (Supplementary Fig. 3e). We also observed enlarged cell size, enlarged nucleus, and increased apoptosis of hepatocytes in *Mettl3* cKO mice starting from 2 weeks and expansion of ductular cells at 3 weeks after birth (Fig. 3a-d and Supplementary Fig. 3f, g). In addition, prominent fibrosis in *Mettl3* cKO livers was seen at 4 weeks and became more pronounced at 5 weeks (Fig. 4a-c and Supplementary Fig. 4a, b). We did not observe any abnormalities of heterozygous cKO livers by histological analysis (Supplementary Fig. 4c). Consistent with the expansion of ductular cells, we detected marked increases of Sox9, CK19, and Ki67 positive cells in *Mettl3* cKO livers (Fig. 4d and Supplementary Fig. 4d). Meanwhile, RT-qPCR of liver tissues collected at different time points showed that hepatocyte markers (*Albumin* (*Alb*)) decreased, while hepatic progenitor markers (*Afp*, *Krt7*, *Krt19*, *Epcam*, and *Sox9*) and fibrosis markers (*Col1a1*, *Acta2*, and *Pdgfrb*) increased in *Mettl3* cKO individuals (Fig. 4e). These changes were confirmed by western blot (Fig. 4f). The above results demonstrate that Mettl3 deletion in hepatocytes perinatally leads to hepatocyte injury, activation of progenitor cells, and fibrosis.

### Transcriptome-wide m⁶A-RIP sequencing to identify potential targets of Mettl3

As Mettl3 is the key catalytic subunit of the m⁶A methyltransferase machinery, to gain a comprehensive insight into the molecular mechanisms underlying Mettl3 regulating postnatal liver development, we first quantified the m⁶A levels on mRNAs of the liver tissues from different developmental stages using LC-MS/MS. Global mRNA m⁶A levels of postnatal livers increased after birth, peaked at 2 weeks, and decreased then (Supplementary Fig. 5a). Next, we profiled the genome-wide m⁶A methylation distribution using m⁶A-RIP sequencing of RNAs from five developmental time points (1 day and 1, 2, 4, and 8 weeks after birth) of mouse liver tissues. The distribution of m⁶A modifications was dynamically regulated during different stages of postnatal liver development (Fig. 5a and Supplementary Fig. 5a-d). We identified 15139, 12483, 12615, 10561, and 6806 m⁶A peaks, corresponding to 6330, 5522, 5515, 4856, and 3445 genes from the above five groups, respectively (Supplementary Fig. 5b and Supplementary Dataset 1). Global m⁶A peak enrichment peaked 2 weeks after birth, along with patterns of bulk m⁶A levels during mouse liver development (Supplementary Fig. 5a, d). In line with previous reports, m⁶A peaks were significantly enriched in the vicinity of the stop codon (Fig. 5a, b), and the consensus motif "GGAC" was most commonly enriched in

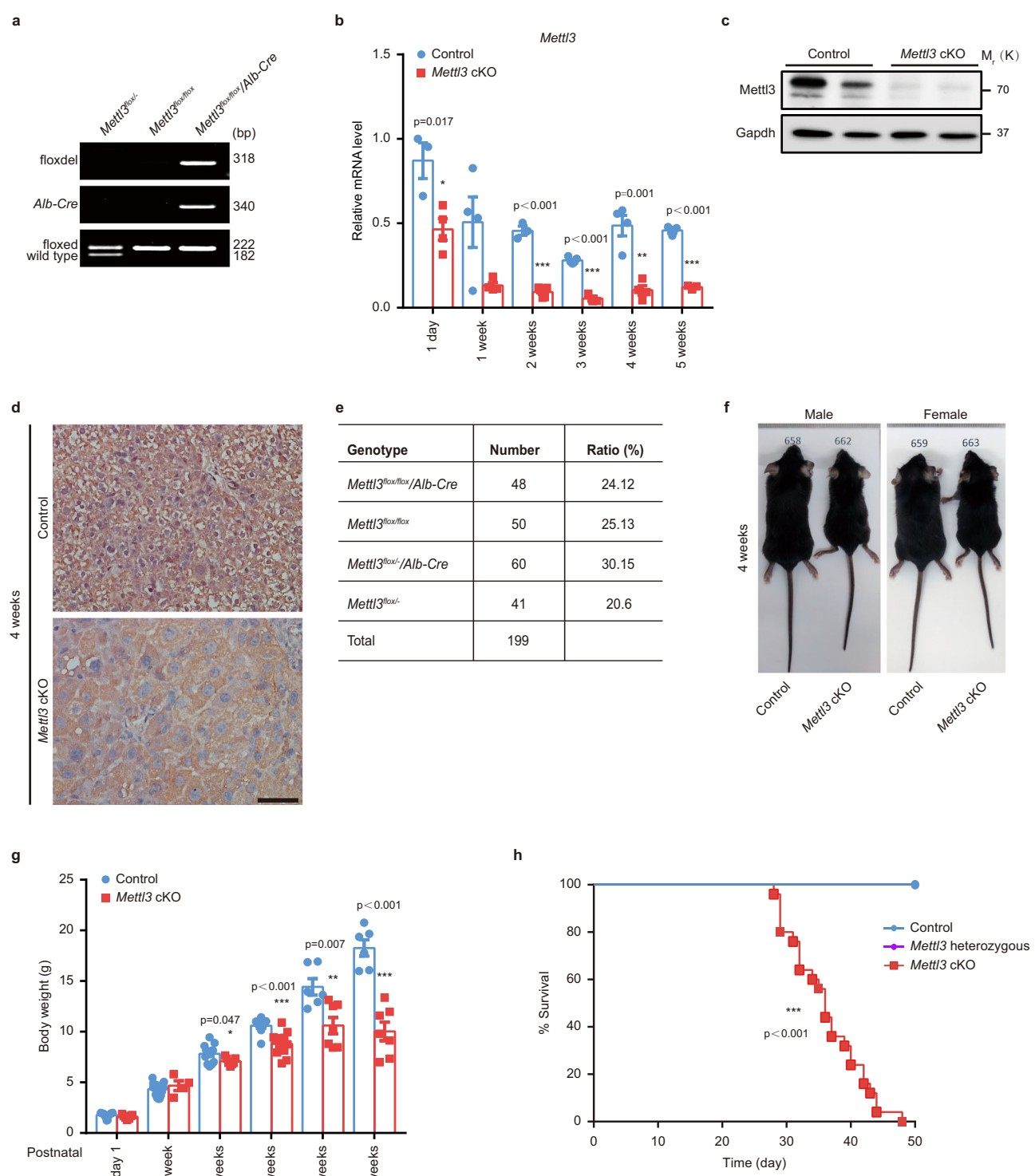

peaks from all samples (Supplementary Fig. 5e). Interestingly, "liver development" was one of the most significantly enriched terms of m[6]A modified genes by Gene Ontology (GO) analysis at all time points (Fig. 5c and Supplementary Dataset 2). mRNAs of several key liver-enriched transcription factors were highly methylated by m[6]A in liver tissues, including *Hnf4a*, *Hnf1a*, *Ppara*, and *Cebpa* (Fig. 5d and Supplementary Fig. 5f). These factors play essential roles in both liver development in vivo and hepatocyte differentiation in vitro[2]. Next, we used gene-specific m[6]A-RIP-qPCR assays to confirm the authentic deposition of RNA m[6]A modifications by Mettl3 on liver-enriched factors using liver tissues from Control and *Mettl3* cKO mice. We

observed a significant decrease of m[6]A deposition on *Hnf1a*, *Hnf4a*, *Ppara*, *Cebpa et al*. in *Mettl3* cKO mice at both 2 weeks and 4 weeks (Fig. 5e). These results indicate that Mettl3-mediated m[6]A modification is dynamically regulated during liver development and modifies crucial transcription factors controlling liver specification and function.

## m[6]A regulates pathways of liver development and metabolism by controlling mRNA stability of the core transcription factor *Hnf4a*

To gain further insights into the mechanism of Mettl3 regulating liver development, we conducted RNA-sequencing for liver tissues from

**Fig. 1 | Hepatic Mettl3 knockout in mice results in postnatal lethality.**
**a** Genomic PCR characterization for tail and liver tissues (the top lane for liver tissues and the rest for tails) from 2 weeks old mouse of the indicated genotype. The top lane (floxdel) showed the exon 2–4 deleted alleles (amplified using F1 and R2 primer shown in (Supplementary Fig. 1c)) that could be detected only in *Mettl3*[flox/flox]/*Alb-Cre* (*Mettl3* cKO) mouse livers. The middle lane (*Alb-Cre*) showed the effective insertion of *Albumin* enhancer/promoter-driven *Cre* into the "genomic safe harbor" Hipp11 (H11) locus. The bottom lane displayed genotyping of heterozygous (*Mettl3*[flox/-]) or homozygous (*Mettl3*[flox/flox]) flox flanking alleles (amplified by F1 and R1 primer shown in (Supplementary Fig. 1c)) (3 experiments were repeated independently with similar results). **b** Quantitation of *Mettl3* mRNA expression in livers of wild-type (WT) control (Control, also hereafter in similar experiments) and *Mettl3* cKO mice at different time points postnatally via RT-qPCR (*n* = 3 for 1 day Control and 3 weeks cKO group; n = 5 for 3 weeks Control group; *n* = 4 for other groups). **c** Western blot for Mettl3 in Control and *Mettl3* cKO mouse liver tissues at two weeks after birth (6 experiments were repeated independently with similar

results). Gapdh was used as a loading control (also hereafter in similar experiments). **d** Immunohistochemistry staining of Mettl3 in 4 weeks old Control and *Mettl3* cKO mouse livers (6 experiments were repeated independently with similar results). Scale bar = 50 µm. **e** The number of offspring with different genotypes from intercrossing *Mettl3*[flox/flox] and *Mettl3*[flox/-]/*Alb-Cre* mice. **f** Representative appearance of sex-matched Control mice and *Mettl3* cKO littermates at 4 weeks after birth. **g** Body weight of male Control and *Mettl3* cKO littermates at different time points after birth (*n* = 4 for 1 week cKO group; *n* = 6 for 4 weeks and 5 weeks Control groups; *n* = 9 for 3 weeks Control group; *n* = 12 for 3 weeks cKO group; n = 13 for 2 weeks Control group; *n* = 15 for day 1 and 1 week Control group; *n* = 7 for other groups). **h** Survival curves of Control, *Mettl3* cKO, and *Mettl3* heterozygous (*Mettl3*[flox/-]/*Alb-Cre*) littermates (n = 25 for each group). Data in **b** and **g** were shown as mean ± SEM with the indicated significance (*$P < 0.05$, **$P < 0.01$, ***$P < 0.001$; two-tailed student's *t*-test). Data in (h) were analyzed by Log-rank (Mantel-Cox) test with the indicated significance (***$P < 0.001$). Source data are provided as a Source Data file.

Control and *Mettl3* cKO mice at 1 day, 1 week, 2 weeks, and 4 weeks after birth. There were much more differentially regulated genes (DEGs) between Control and *Mettl3* cKO mice at later time points (Supplementary Fig. 6a, Supplementary Dataset 3), which is consistent with our observations that *Mettl3* cKO mice showed progressive severe liver damage 2 weeks after birth onward (Fig. 3a). Gene set enrichment analysis (GSEA) showed that targets of *Hnf4a* and *Hnf1a* were significantly repressed in *Mettl3* cKO livers even at 1 day postnatally (Fig. 6a and Supplementary Fig. 6b). Dual-luciferase reporter assay and mutagenesis assay (Fig. 6b and Supplementary Fig. 6c, d) showed that co-transfection with WT, but not catalytic mutant Mettl3[22–24], significantly promoted luciferase activity in reporters carrying WT *Hnf4a* and *Hnf1a* fragments, while such increases were abolished when the m[6]A consensus motifs were mutated, confirming that the regulation of Hnf4a and Hnf1a by Mettl3 was indeed relying on m[6]A methylation of their transcripts. Both RT-qPCR and western blot confirmed that Hnf4a was downregulated in *Mettl3* cKO livers at different time points postnatally (Fig. 6c-e). Although the RNA level of *Hnf1a* was downregulated at all time points (Supplementary Fig. 6e), we observed a dramatic decrease of Hnf1a protein with age and only observed a difference between Control and *Mettl3* cKO mouse livers 1 week after birth (Supplementary Fig. 6f), indicating a less essential role of Hnf1a in the maturation of hepatocytes, which is consistent with previous studies[25]. Since Hnf4a is a master transcription factor required for liver development in both foetuses and adults and controls most aspects of mature hepatocyte function[26,27], we mainly focused on Hnf4a for further studies. RNA-sequencing data showed that along with the downregulation of Hnf4a, most Hnf4a target genes, such as *Apoa2, Apoc3, Cyp8b1,* and *Mttp,* were repressed in *Mettl3* cKO individuals (Supplementary Fig. 6g, Supplementary Dataset 3), which was validated by RT-qPCR (Supplementary Fig. 6h). We also noticed that *Smad* signaling, the central mediator of fibrosis[28], was significantly enriched in *Mettl3* cKO mouse liver tissues at 4 weeks (Supplementary Fig. 7a), supporting the phenomenon that massive liver fibrosis was induced in *Mettl3* cKO animals (Fig. 4). These results indicate that Mettl3-mediated m[6]A controls the expression of crucial liver developmental genes during liver development.

m[6]A modification is involved in various aspects of RNA metabolism, including transcription, splicing, nuclear transportation, stability, and translation. Because we observed decreased expression of Hnf4a at both mRNA and protein levels, we determined the alternative splicing, nucleus-cytoplasm transportation, and mRNA stability of *Hnf4a* mRNA. Alternative splicing analysis showed no differences on *Hnf4a* transcripts in RNA-sequencing data from Control and *Mettl3* cKO livers (Supplementary Dataset 4). The distribution of *Hnf4a* mRNA in nuclear and cytoplasm was also not affected by *Mettl3* knockout (Supplementary Fig. 7b-d). Only mRNA stability showed significant changes in primary hepatocytes and the HepG2 cells with Mettl3 inhibition

(Fig. 6f, g, and Supplementary Fig. 7e, f). Cells with Mettl3 deletion showed a shorter half-life of *Hnf4a* transcript, suggesting that Mettl3-mediated m[6]A controls the expression of Hnf4a at least partly by regulating its mRNA stability. To compare the global changes of mRNA stability after Mettl3 knockout, we subjected actinomycin D-treated hepatocytes from Control and *Mettl3* cKO mice for RNA-sequencing (Supplementary Dataset 5). Consistent with previous reports[15,19], knockout of Mettl3 enhanced mRNA stability globally, especially for m[6]A-modified genes (Supplementary Fig. 7g, h). Among genes involved in liver development (defined by Gene Ontology Resource, GO:0001889), only *Hnf4a* and another 10 genes showed decreased mRNA half-life when Mettl3 was knocked out, while most genes (including *Cited2, Cebpa, Notch2, Dbp,* et al.) were more stable or unchanged (Supplementary Fig. 7i and Supplementary Dataset 5). These results demonstrated that Mettl3 deficiency downregulated Hnf4a expression by reducing the half-life of *Hnf4a* mRNA.

m[6]A modification controls RNA fate mainly through "reader" proteins recognizing and binding to m[6]A-containing transcripts. Among identified m[6]A "readers", insulin-like growth factor 2 mRNA-binding proteins (IGF2BPs, including IGF2BP1/2/3) are known to promote the stability of their target mRNAs[17]. To further delineate the mechanism of m[6]A controlling Hnf4a expression, we checked previous publications and found that deletion of IGF2BP1 leads to destabilization of *HNF4A* mRNA in HepG2 cells while interfering with the other two members did not affect *HNF4A* mRNA degradation (Supplementary Fig. 7j)[17], indicating that IGF2BP1 may directly recognize m[6]A on *HNF4A* mRNA and maintain its levels in the liver context. Thus, we tested Igf2bp1 binding on *Hnf4a* mRNA with RIP experiments. The results showed that Igf2bp1 could efficiently bind to *Hnf4a* transcripts in mouse livers of both 2 weeks and 4 weeks, and the enrichment significantly decreased after *Mettl3* knockout (Fig. 6h). Accordingly, knocking down IGF2BP1 with small Hairpin RNA (shRNA) in HepG2 cells also decreased *HNF4A* mRNA half-life, similar to Mettl3 disruption (Fig. 6i and Supplementary Fig. 7k, l). These data demonstrated that Mettl3-mediated m[6]A controls the expression of Hnf4a by regulating its mRNA stability in an IGF2BP1-dependent manner.

## Hepatic Hnf4a overexpression alleviated liver injury caused by Mettl3 knockout

To further strengthen our conclusion that Hnf4a is the primary mediator of Mettl3 function in liver development, we conducted rescue experiments using AAV serotype 8 (AAV8) to express Hnf4a under the control of a liver-specific promoter (thyroxine-binding globulin, TBG) (AAV8-TBG-Hnf4a) on *Mettl3* cKO mice (Fig. 6j). Injection of AAV8-TBG-Hnf4a by superficial temporal vein on day two after birth successfully overexpressed Hnf4a in the liver (Supplementary Fig. 7m) and alleviated liver damage caused by hepatic *Mettl3* knockout compared to AAV8-Ctrl at two weeks, evidenced by an increased number of

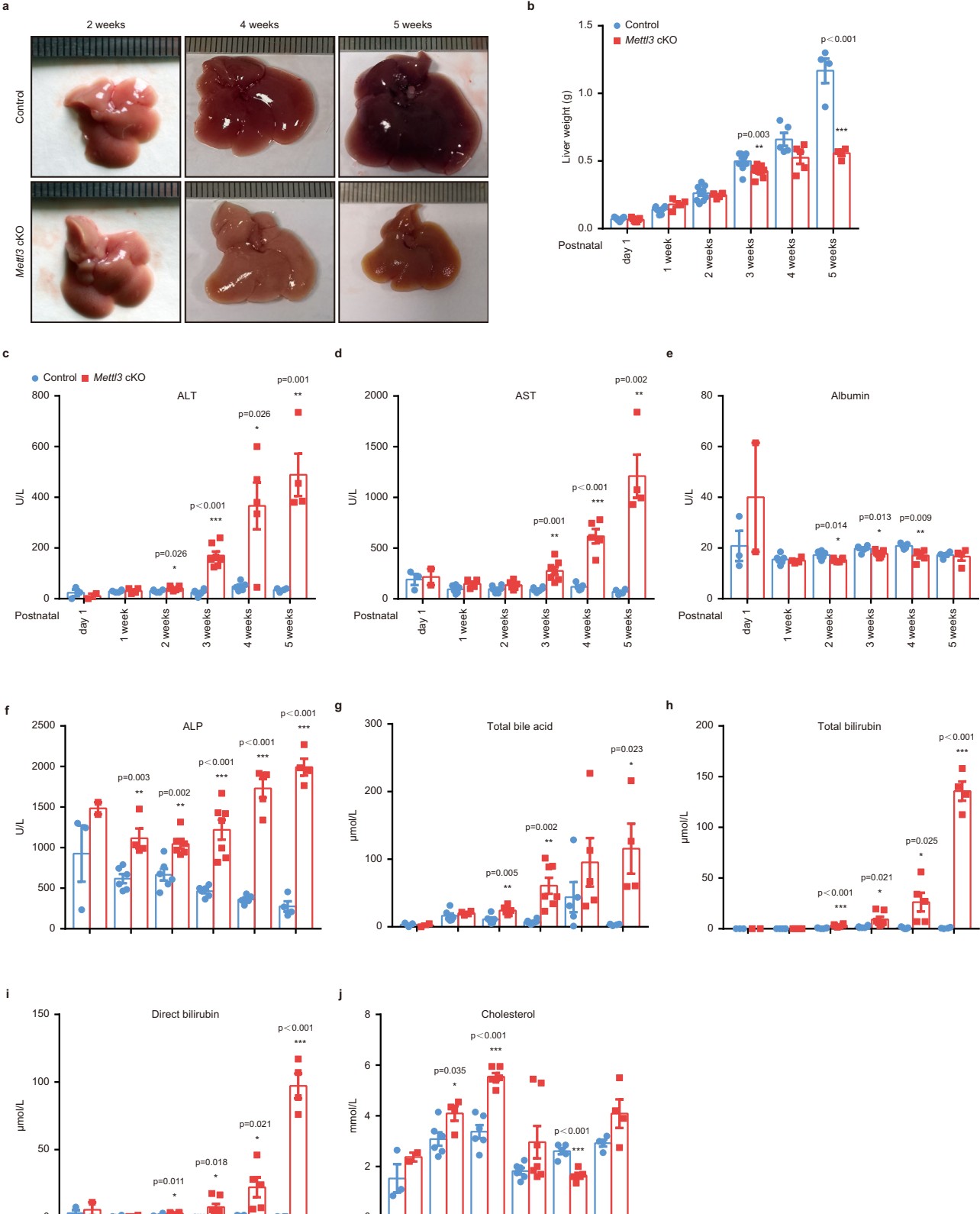

**Fig. 2 | Hepatic Mettl3 knockout in mice causes liver injury. a** Representative gross appearance of livers from Control and *Mettl3* cKO mice at 2 weeks, 4 weeks, and 5 weeks postnatally. **b** Liver weight at different time points postnatally (*n* = 4 for 1 week cKO group and 5 weeks groups; *n* = 5 for 4 weeks groups; *n* = 6 for 2 weeks cKO group; *n* = 7 for day 1 cKO group; *n* = 8 for 1 week Control group; *n* = 9 for 3 weeks Control group; *n* = 10 for 2 weeks Control group; *n* = 12 for 3 weeks cKO group; *n* = 15 for day 1 Control group). **c**–**j** Serum levels of ALT **c**, AST **d**, Albumin **e**, ALP **f**, Total bile acid **g**, Total bilirubin **h**, Direct bilirubin **i**, and Cholesterol **j** of Control and *Mettl3* cKO mice at different time points postnatally (*n* = 2 for day 1 cKO group; *n* = 3 for day 1 Control group; *n* = 4 for 1 week cKO and 5 weeks groups; *n* = 5 for 4 weeks groups; *n* = 6 for 1 week Control group, 2 weeks groups, and 3 weeks Control group; *n* = 7 for 3 weeks cKO group). Data in **b**–**j** were shown as mean ± SEM with the indicated significance (*P < 0.05, **P < 0.01, ***P < 0.001, two-tailed student's *t*-test). Source data are provided as a Source Data file.

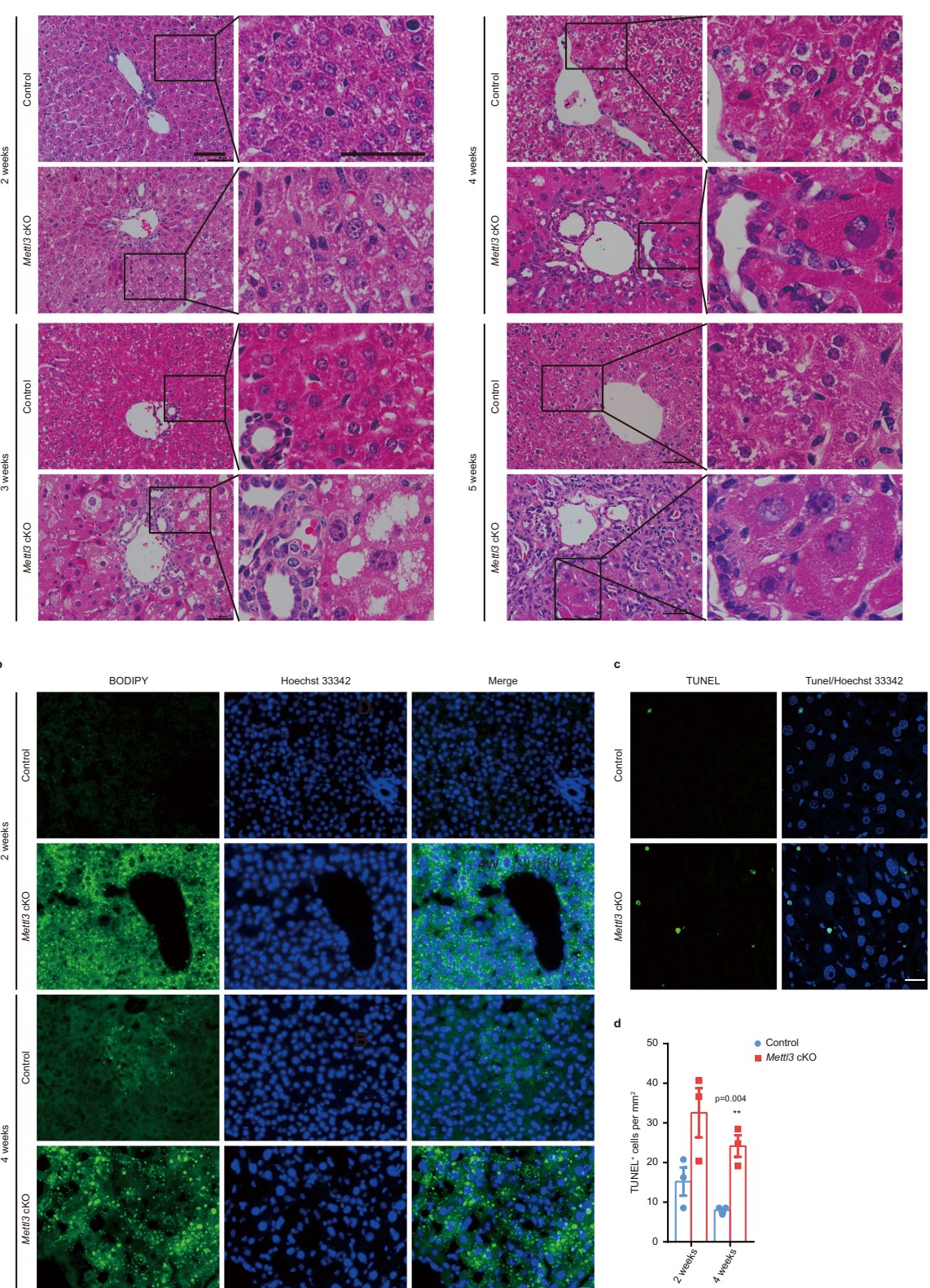

**Fig. 3 | Hepatic deletion of Mettl3 in mice induces steatosis and apoptosis.**
**a** Representative H&E staining photographs of liver sections from Control and *Mettl3* cKO mice at indicated time points postnatally (10 experiments were repeated independently with similar results). Scale bar = 50 μm. **b** Representative BODIPY staining fluorescent photographs of frozen liver sections from Control and *Mettl3* cKO mice at 2 weeks and 4 weeks after birth (6 experiments were repeated independently with similar results). Cell nuclei were counterstained with Hoechst 33342 (also hereafter in similar experiments). Scale bar = 20 μm. **c** Representative

TUNEL staining fluorescent photographs for frozen liver sections from Control and *Mettl3* cKO mice 4 weeks after birth (6 experiments were repeated independently with similar results). Scale bar = 20 μm. **d** Quantification of TUNEL[+] cells/Hoechst 33342[+] cells ratio in liver sections from Control and *Mettl3* cKO mice at 2 weeks and 4 weeks after birth (n = 3 for each group). Data in **d** were shown as mean ± SEM with the indicated significance (**P < 0.01; two-tailed student's *t*-test). Source data are provided as a Source Data file.

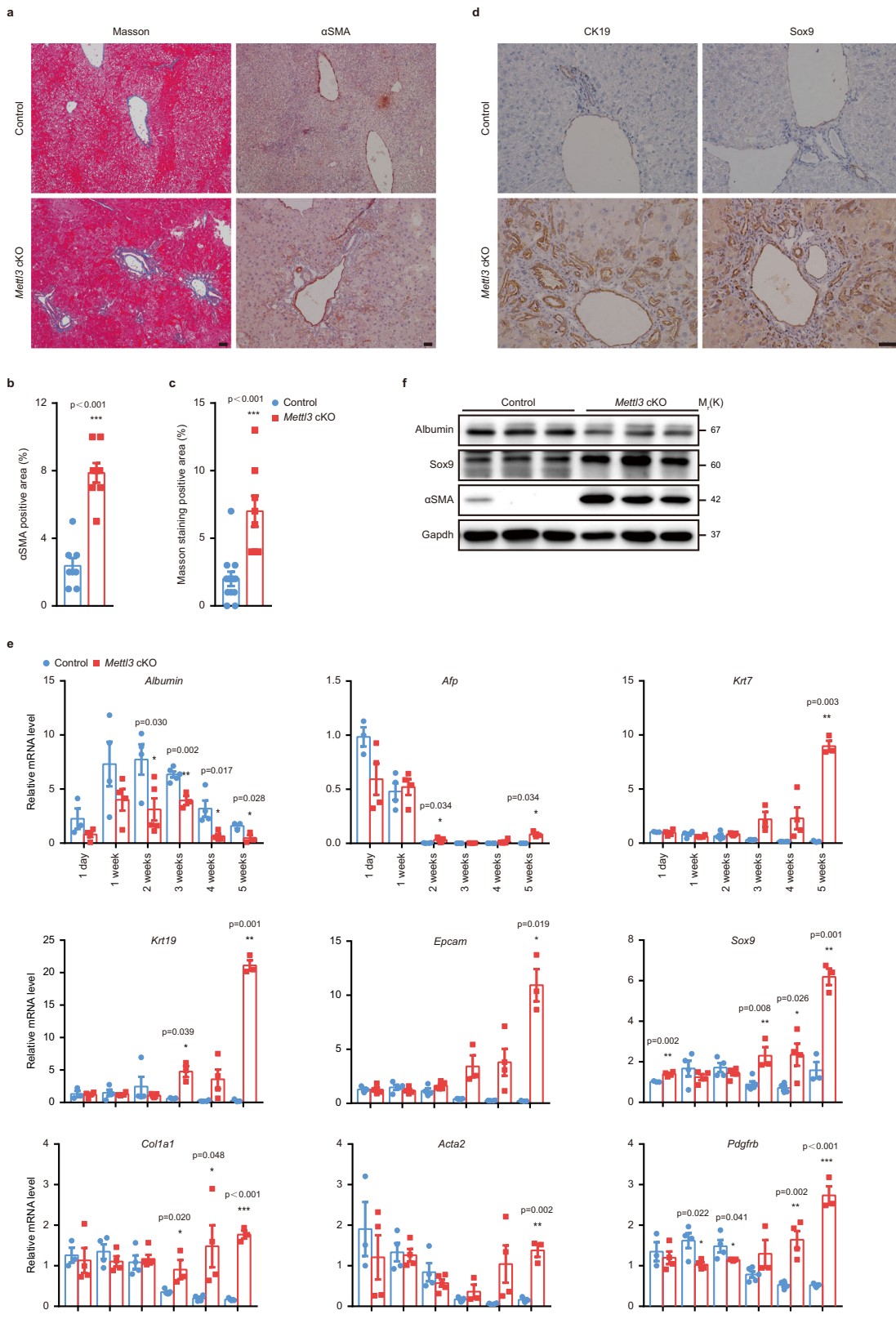

Ki67[+] proliferating hepatocytes and reduced hepatic steatosis (Fig. 6k-m). However, we did not see long-term benefits on mortality. This may attribute to the rapid dilution of AAV caused by the vigorous hepatocyte division within four weeks after birth[29]. Then we overexpressed Hnf4a by AAV-TBG-Hnf4a through tail vein injection at four-week-old *Mettl3* cKO mice and found that Hnf4a overexpression significantly prolonged the life span of *Mettl3* cKO mice (Fig. 6n). These results

further demonstrated that Hnf4a is the primary factor mediating the function of Mettl3 in liver development.

## Mettl3 is dispensable for homeostasis of adult liver

Given the lethality of *Mettl3* cKO mice within 7 weeks, we generated conditional inducible *Mettl3* knockout mice (*Mettl3* icKO) by crossing *Mettl3*^flox/flox mice with *Alb-Cre*^ERT2 mice. Mettl3 could be deleted in adult

**Fig. 4 | Hepatic deletion of Mettl3 in mice induces fibrosis and activation of hepatic progenitors. a** Representative Masson's trichrome staining and αSMA immunohistochemistry staining photographs of liver sections from Control and *Mettl3* cKO mice at 4 weeks after birth (10 experiments were repeated independently with similar results). Scale bar = 50 μm. **b** Quantification of the αSMA positive area (*n* = 8 for each group). **c** Quantification of the Masson's trichrome staining positive area (*n* = 8 for cKO group; *n* = 12 for Control group). **d** CK19 and Sox9 immunohistochemistry staining of liver sections from Control and *Mettl3* cKO mice at 4 weeks after birth (6 experiments were repeated independently with similar results). Scale bar = 50 μm. **e** RT-qPCR analysis of hepatocyte markers, hepatic progenitor markers, and fibrosis markers for liver tissues from Control and *Mettl3* cKO mice at different time points postnatally (*n* = 3 for 1 day Control group, 3 weeks cKO group, and 5 weeks groups; *n* = 5 for 3 weeks Control group; *n* = 4 for other groups). **f** Western blot for Albumin, Sox9, and αSMA of Control and *Mettl3* cKO mouse liver tissues at 4 weeks after birth (3 experiments were repeated independently with similar results). Data in **b**, **c**, and **e** were shown as mean ± SEM with the indicated significance (*$P < 0.05$, **$P < 0.01$, ***$P < 0.001$; two-tailed student's *t*-test). Source data are provided as a Source Data file.

mice by intraperitoneal (IP) injection of tamoxifen[30]. Genomic PCR, RT-qPCR, and western blot confirmed efficient and specific depletion of *Mettl3* 1 week after tamoxifen administration (Fig. 7a-e and Supplementary Fig. 8a). LC-MS/MS results also showed a significant decrease in bulk m6A modification of *Mettl3* icKO mouse liver mRNAs (Fig. 7f). However, we did not observe any visible abnormalities in these *Mettl3* icKO mice (Fig. 7g and Supplementary Fig. 8b, c). Serological and histologic examinations showed minimal liver damage (Fig. 7h, i, and Supplementary Fig. 8d). Both mRNAs of *Hnf4a* and downstream targets of Hnf4a showed no differences between Control and *Mettl3* icKO livers (Supplementary Fig. 8e-k). These results indicate that although Mettl3 is essential for the early postnatal development of the liver, it is not crucial for the homeostasis of adult liver (Supplementary Fig. 9).

## Discussion

Previous reports had profiled transcriptome-wide m6A in porcine liver at three postnatal stages[31] and supplied a roadmap of m6A modification across human and mouse livers[32], indicating dynamic changes of m6A modification during liver development. This study demonstrated a vital role of Mettl3-mediated m6A modification in mouse postnatal liver development. Using *Mettl3^flox/flox^/Alb-Cre* mice, we found that *Mettl3* cKO mice got steatosis at about 2 weeks after birth, liver fibrosis at 4–5 weeks, and finally died before 7 weeks due to severe liver injury.

The liver experiences abrupt functional changes from intra- to the extra-uterine environment, corresponding to a functional shift from haematopoiesis to metabolism and immunity[33]. In general, fetal liver haematopoiesis is characterized by initiation (E11.5), peak (E14.5), recession (E15.5), and disappearance (3 days after birth), while the neonatal liver rapidly evolves into a vital organ for immunosurveillance and metabolism. Histological analysis showed that haematopoietic cells disappeared rapidly, and parenchymal cells occupied in hepatic constituent in the first week after birth, consistent with previous reports[34]. Mettl3 was barely expressed in neonatal mouse livers, while highly expressed at both fetal livers[35] and 1 week after birth at protein levels (even though the mRNA level only slightly changed during this process). The mechanism of how Mettl3 is tightly regulated during this perinatal period remains elusive and is worth considering. Given the functional transition of livers perinatally, the dynamic regulation and function of Mettl3 in this period might be involved in both haematopoietic and hepatic aspects. A series of studies have investigated the role of Mettl3-mediated m6A in haematopoietic system[36,37], including early haematopoietic stem cell (HSC) development in the fetal liver[35], revealing an essential role for m6A in both specification and homeostasis of haematopoietic system. However, the function of Mettl3 and m6A in hepatic lineage specification in prenatal development is currently not clear and worthy to be studied further.

Interestingly, the expression of Mettl3 and Mettl14 decreased synchronously in the liver after 4 weeks old, raising the possibility of a less critical role of m6A modification in adulthood and elder age. Indeed, we found that Mettl3 was not essential for liver homeostasis in adults. Previous studies also showed higher expression levels of the methyltransferase complex in early stages of differentiation or progenitors, such as in neurogenesis[38,39] and haematopoiesis[40]. The level of global m6A modification and methyltransferase expression is

reduced in premature mesenchymal stem cells[41], replicative senescent cells[42], and peripheral blood mononuclear cells from old cohorts[43]. Aging is always accompanied by a progressive decline in regenerative capacity, especially in the liver[44], and thus it will be intriguing to delineate whether the age-related decline of regeneration capacity is elicited by decreased m6A dynamics in older individuals. Although overexpressing RNA methyltransferases attenuates the senescence phenotype[41–43], increased methyltransferase expression is related to aggravated liver metabolic disorders[45] and carcinogenesis progression[46], including hepatocellular carcinoma[47], indicating that fine-tuning regulation of m6A machinery is essential for physiological homeostasis.

As the largest digestive and metabolic organ in adults, the liver is responsible for transforming protein, glycogen, cholesterol, fatty acid, and many other complex molecules into elementary molecules. The proper development and function of the liver were maintained by a series of liver-enriched transcription factors and comprehensive regulatory networks among them[2,48]. Here we found that plenty of liver-enriched transcription factor transcripts were modified by Mettl3-mediated m6A modification, including *Hnf1a*, *Hnf1β*, *Hnf4a*, *Ppara*, *Cebpa*, *Onecut1*, *Onecut2*, *Cited2*, et al. *Hnf4a* seemed to be the most critical downstream mediator of Mettl3 in postnatal liver development. Furthermore, we observed a decreased Hnf4a at different time points on both mRNA and protein levels in *Mettl3* cKO mouse livers compared with Control livers and significant dysregulation of its downstream targets.

Hnf4a is a master transcription factor required for mouse livers in both foetuses and adults[26]. It directly binds to almost half of the actively transcribed genes in adult livers and serves as a high-level transcription factor in hepatic transcriptional hierarchies[49,50]. Most of the abnormalities we found in *Mettl3* cKO mice (including lipid deposition in the liver, increased bile acids in serum, liver injury, and lethality in young adults) phenocopied that of hepatic *Hnf4a* knockout mice[26]. However, *Mettl3* cKO mice showed more severe liver injury than hepatic *Hnf4a* knockout mice. This might be explained by the fact that multiple targets were regulated by Mettl3-mediated m6A modification, and auto-regulatory and cross-regulatory circuits among them may further accelerate the collapse of liver development and function[49,51,52]. Among others, accumulating evidence showed profound crosstalk between m6A and histone/DNA epigenetic modifications[53], which added another layer of complexity to explain the role of Mettl3 in liver development and function. Recent studies showed that m6A directly regulates heterochromatin organization[54,55]. Coordinated remodeling of heterochromatin is essential for liver development[56], and disturbed heterochromatin landscape contributes to impaired hepatic function and tumorigenesis[57]. Therefore, it will be interesting to see whether an abnormal chromatin state accounts for *Mettl3* cKO-induced liver injury. Besides, Mettl3 knockout may also lead to liver development defects by controlling other aspects of RNA metabolism, such as mRNA transportation, translation, and splicing of other regulators involved in liver development and biogenesis of miRNAs critical for hepatogenesis.

In summary, our study demonstrated a novel regulatory function of epitranscriptomics by Mettl3-mediated m6A modification on liver postnatal development and homeostasis, expanding our

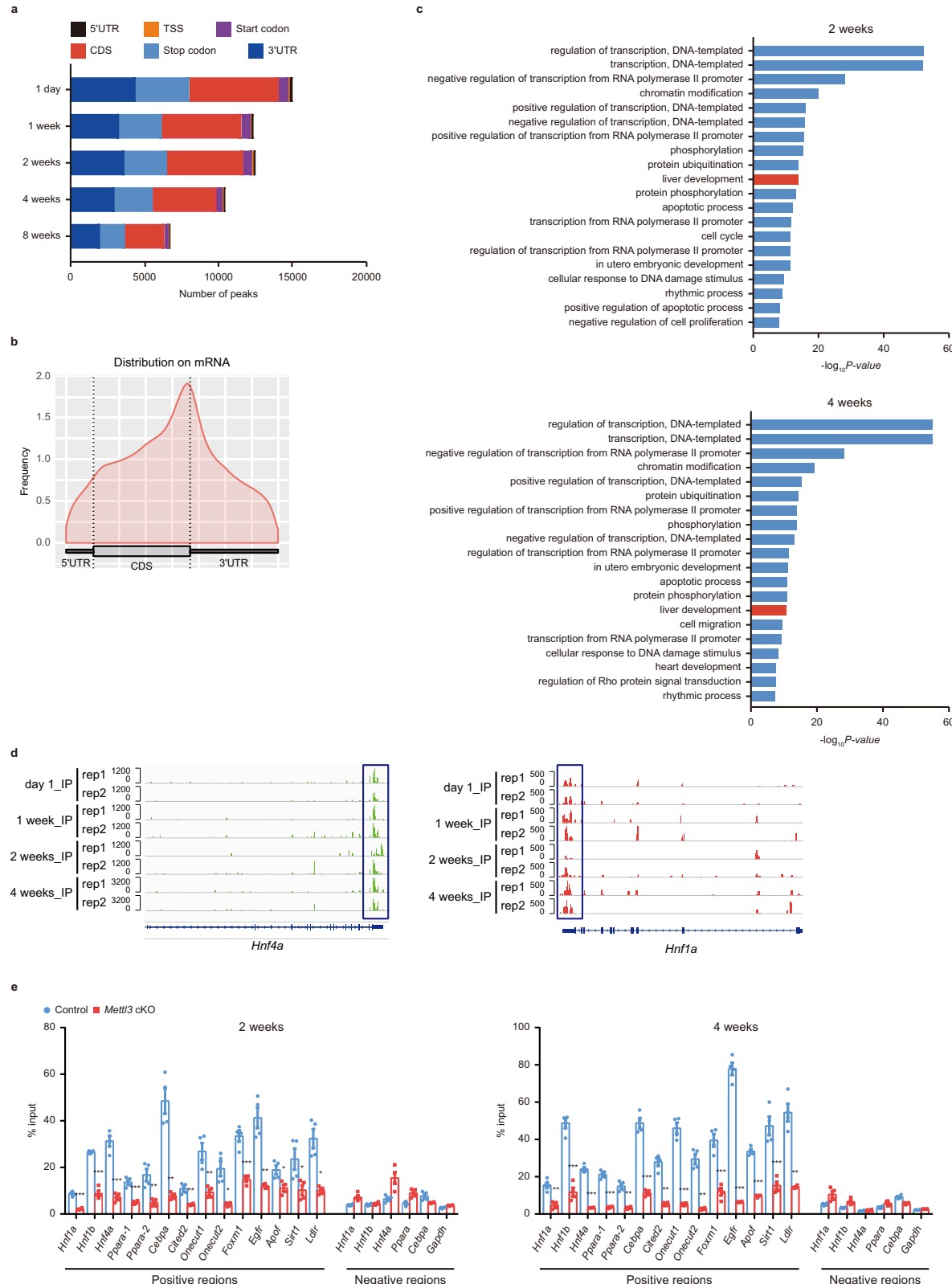

understanding of the regulatory network in mammalian liver development and function.

## Methods

### Ethics statement

Animals in this study were used in accordance with the Guide for Care and Use of Laboratory Animals of the National Institute of Health, and protocols were approved by the Institutional Animal Care and Use Committee of The Third Affiliated Hospital of Sun Yat-sen University. For human specimens used in this study, informed consent was obtained from all individuals and recorded in the electronic database. Research Board protocols were conducted under the guidelines set out by the Medical Ethical Committees of the Third Affiliated Hospital of Sun Yat-sen University.

**Fig. 5 | Mettl3-mediated m⁶A modification regulates postnatal liver development dynamically. a** Summary of peak numbers and the distribution of m⁶A peaks in the 5′UTR, TSS, start codon, CDS, stop codon, and 3′UTR regions across the entire set of mRNA transcripts. **b** Metagene distribution of m⁶A-immunoprecipitated reads across the length of mRNA transcript of livers from mice at postnatal 2 weeks. **c** Gene ontology (GO) analysis of m⁶A modified genes in mouse liver tissues at 2 and 4 weeks after birth identified by m⁶A-RIP sequencing. **d** Genome Browser screenshots of m⁶A-RIP sequencing read density signals under 4 different time points after birth at *Hnf4a* and *Hnf1a* loci. Two replicates were contained at each time point. **e** Gene-specific m⁶A-RIP-qPCR for Control and *Mettl3* cKO mouse liver tissues at 2 and 4 weeks after birth (*n* = 4 for each group over 3 independent experiments). m⁶A positive and negative regions were selected based on m⁶A-RIP sequencing. Genes were

marked by serial number if two or more m⁶A positive regions were found in an individual gene (e.g., *Ppara*−1 and *Ppara*−2 meant two separate m⁶A positive regions in *Ppara* mRNA). Exact P-value for each gene between Control and cKO in 2 weeks positive regions were listed here: *Hnf1a* (<0.001), *Hnf1b* (<0.001), *Hnf4a* (<0.001), *Ppara*-1 (<0.001), *Ppara*-2 (0.005), *Cebpa* (0.004), *Cited2* (0.009), *Onecut1* (0.004), *Onecut2* (0.016), *Foxm1* (<0.001), *Egfr* (0.007), *Apof* (0.018), *Sirt1* (0.040), *Ldlr* (0.011). Exact P-value for each gene between Control and cKO in 4 weeks positive regions were listed here: *Hnf1a* (0.001), *Hnf1b* (<0.001), *Hnf4a* (<0.001), *Ppara*-1 (<0.001), *Ppara*-2 (0.003), *Cebpa* (<0.001), *Cited2* (0.001), *Onecut1* (<0.001), *Onecut2* (0.002), *Foxm1* (<0.001), *Egfr* (<0.001), *Apof* (<0.001), *Sirt1* (<0.001), *Ldlr* (0.003). Data in (e) were shown as mean ± SEM with the indicated significance (*$P < 0.05$, **$P < 0.01$, ***$P < 0.001$; two-tailed student's *t*-test). Source data are provided as a Source Data file.

## Animal experiments

*Mettl3^flox/flox^* mice were kindly gifted from Professor Qi Zhou[58]. *Alb-Cre* mice were purchased from the GemPharmatech Co. Ltd (Nanjing, China). *Alb-Cre^ERT2^* mice were purchased from Beijing Biocytogen Co., Ltd. (Beijing, China). All the mice were C57BL/6 J background and housed in a specific pathogen-free facility under 12 h light/dark cycle, a temperature of 24 ± 2 °C, humidity between 30 and 70%, with access to food and water ad libitum. *Mettl3^flox/flox^*/*Alb-Cre* mice (*Mettl3* cKO mice) were generated by crossing *Mettl3^flox/flox^* mice with heterozygous *Alb-Cre* mice. *Mettl3^flox/flox^*/*Alb-Cre^ERT2^* mice were generated by crossing *Mettl3^flox/flox^* mice with *Alb-Cre^ERT2^* mice. For liver-specific inducible knockout of *Mettl3* (*Mettl3* icKO) in adults, 4-5 weeks old *Mettl3^flox/flox^*/ *Alb-Cre^ERT2^* mice were treated with tamoxifen (Sigma-Aldrich, T5648) at 1 mg/mouse for 5 consecutive days by intraperitoneal injection, and control mice were treated with the same volume of olive oil (MACKLIN, O815211). Tamoxifen was dissolved in olive oil at a concentration of 10 mg/mL by shaking overnight at 37 °C. Control and *Mettl3* cKO or *Mettl3* icKO mice were littermates and cage mates. All the *Alb-Cre* or *Alb-Cre^ERT2^* mice in this paper were heterozygous for Cre.

## Genomic PCR for mouse genotyping

Mice were genotyped with tail and tissue DNA. Tail and tissue lysates were prepared using a Mouse Direct PCR kit (ApexBio Technology, k1025). Two pairs of primers were used to identify floxed alleles: *Mettl3*-F1 and *Mettl3*-R1, or *Mettl3*-F2 and *Mettl3*-R2 (shown in Supplementary Fig. 1c). *Mettl3*-F1 and *Mettl3*-R1 primer (shown in Supplementary Fig. 1c) were used to distinguish WT (182 bp) or floxed allele (222 bp) with mouse tails. *Mettl3*-F2 and *Mettl3*-R2 primer (shown in Supplementary Fig. 1c) were used to distinguish WT (295 bp) or floxed allele (335 bp) with mouse tails. *Alb-Cre*-F and *Alb-Cre*-R primer were used to detect the *Alb-Cre* allele with mouse tails (340 bp for *Alb-Cre^+/-^* and none for *Alb-Cre^-/-^*). *Alb-CreERT*-F and *Alb-CreERT*-R primer were used to detect the *Alb-Cre^ERT^* allele with mouse tails (788 bp for *Alb-Cre^ERT2+/-^* and none for *Alb-Cre^ERT2 -/-^*). For floxdel detection in different tissues, *Mettl3*-F1 and *Mettl3*-R2 primers were used. A 318 bp band of floxdel could be only observed in tissues with successful deletion of *Mettl3*, accompanied by a much thinner WT band at about 2500 bp. Uncropped gels were provided in the Source Data file. Detailed primer sequences were listed in Supplementary Table 1.

## AAV virus preparation and in vivo transduction

Serotype 8 AAV (AAV8) was used in this study. Hnf4a coding sequencing was cloned into an AAV8 vector under the control of TBG promoter to achieve liver-specific overexpression of Hnf4a protein. The virus was packaged by Packgene Biotech Co., Ltd (Guangzhou, China) with a final titer larger than $2.0 \times 10^{13}$ viral genomes/mL. For AAV transduction in vivo, we used two strategies for rescue experiments. In the first strategy, each *Mettl3* cKO mouse received $1.0 \times 10^{11}$ viral genomes of AAV8-Ctrl or AAV8-Hnf4a through the superficial temporal vein on day 2 after birth. Mice were sacrificed at 2 weeks old for further analysis. In another strategy, $1.0 \times 10^{11}$ viral genomes/mouse of AAV8-

Ctrl or AAV8-Hnf4a was injected into *Mettl3* cKO mice through the tail vein at 4 weeks old for further survival rate analysis.

## Serum analysis

Serum levels of liver function indicators (alkaline phosphatase (ALP), alanine aminotransferase (ALT), aspartate aminotransferase (AST), albumin, cholesterol, triglyceride, total bile acid, total bilirubin, and direct bilirubin) were detected using Hitachi 7020 automatic biochemical analyser (Hitachi, Tokyo, Japan).

## Human specimens

Human liver tissues were obtained from donation after cardiac death (DCD) during liver transplantation in the Third Affiliated Hospital of Sun Yat-sen University. The study has been approved by the Medical Ethical Committees of the Third Affiliated Hospital of Sun Yat-sen University. The study design and conduct complied with all relevant regulations regarding the use of human study participants and was conducted following the criteria set by the Declaration of Helsinki.

## Cell culture

HEK293T and HepG2 cells were obtained from American Type Culture Collection (ATCC) and maintained in DMEM-high glucose medium (Thermo Scientific, C11995500BT) supplemented with 10% fetal bovine serum (FBS) (PAN, P30-3302). Cells were incubated at 37 °C in a humidified atmosphere of 5% CO₂.

## Primary hepatocyte isolation

Primary hepatocytes were isolated according to traditional two-step collagenase perfusion methods[59]. Briefly, mice were perfused through portal vein cannulation by EGTA buffer (8000 mg/L NaCl, 400 mg/L KCl, 76.67 mg/L NaH₂PO₄, 120.45 mg/L Na₂HPO₄, 2380 mg/L HEPES, 350 mg/L NaHCO₃, 190 mg/L EGTA, and 900 mg/L Glucose, PH = 7.35-7.4) at 8 mL/min for 2 min, followed by enzyme buffer (8000 mg/L NaCl, 400 mg/L KCl, 76.67 mg/L NaH₂PO₄, 120.45 mg/L Na₂HPO₄, 2380 mg/L HEPES, 350 mg/L NaHCO₃, and 481.8 mg/L CaCl₂, PH = 7.35-7.4) containing 100 U/mL collagenase IV (Sigma-Aldrich, C5138) at 8 mL/min for 8 min. The cell suspensions were filtered through a 70 μm cell strainer (Sorfa, 251200). Cell pellets were collected by centrifugation at 50 g for 1 min at 4 °C after three washes. Then cells were resuspended in Williams' Medium E (GIBCO, 12551032) supplemented with 10% FBS and 1% penicillin/streptomycin (KeyGEN Biotech, KGY0023) and seeded in Type I Collagen (Invitrogen, A048301) precoated culture plates and cultured at 37 °C in 5% CO₂ incubator. The culture medium was changed at 2 hours of incubation. Cells were changed to serum-free medium 6 hours later and cultured overnight before use.

## Plasmid construction and virus transduction

shRNA targeting human *METTL3* (sh*METTL3*), human *IGF2BP1* (sh*IGF2BP1*), and luciferase (sh*Luc*) were cloned into pLKO.1 lentiviral vector (Addgene, 10878). All constructs were confirmed by Sanger

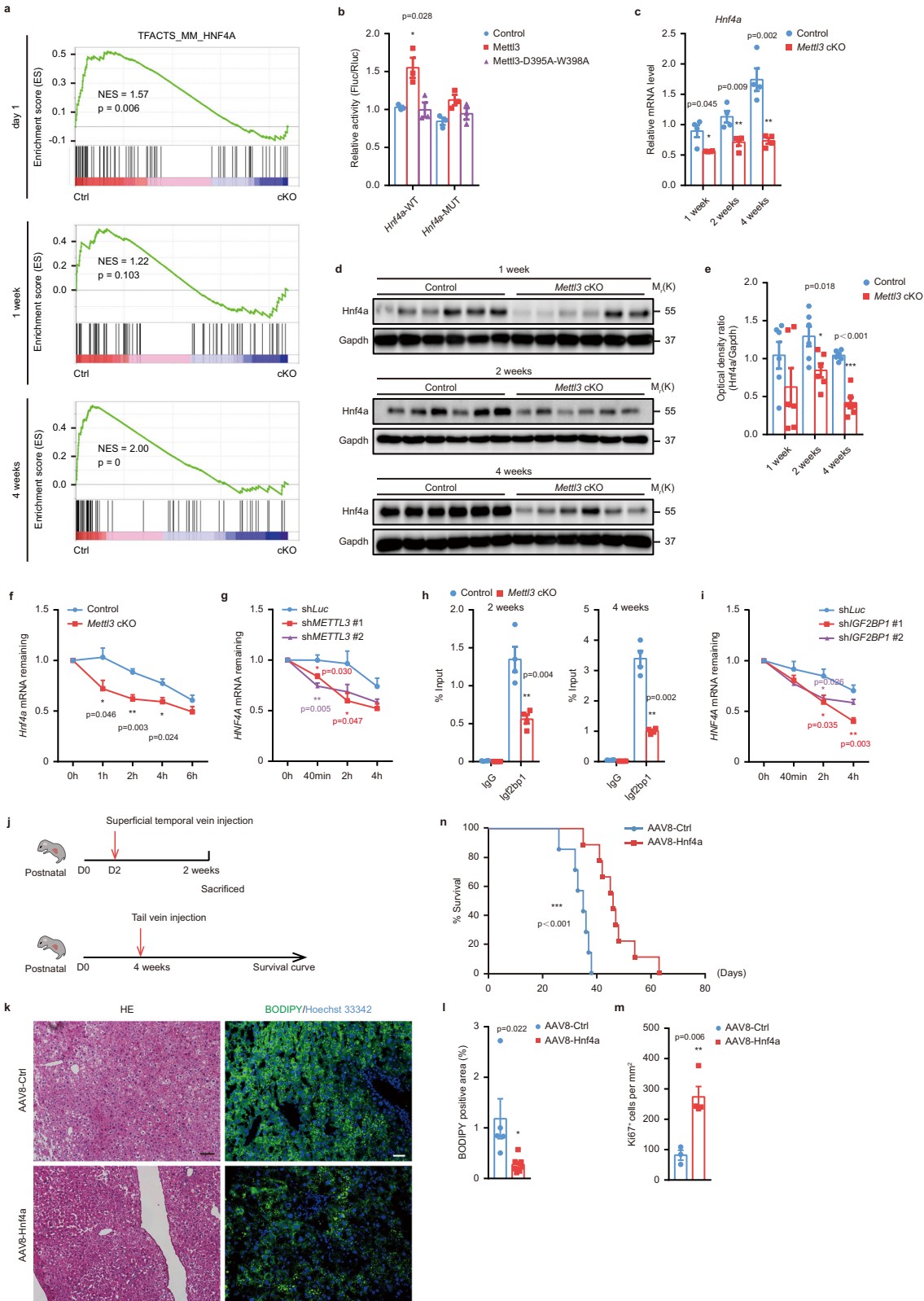

sequencing. Lentivirus transfections were conducted using poly-ethylenimine (PEI, Polysciences, 23966) according to the manu-facturer's protocol. Briefly, HEK293T cell was seeded into 6 cm dish, and transfection experiments were performed at 80% confluence. For each well, 335 μl of pre-warmed Opti-MEM (Invitrogen) was mixed with 8 μg of plasmids (target plasmid: psPAX2: pMD2.G = 4:2:1), 36 μl of PEI

(1 μg/ml), then incubated for 14 min at room temperature and added to the dish. The culture medium was changed 10 hours later. 48 h after transfection, the supernatants were collected and filtered through a 0.45 μm filter. HepG2 cells were infected with lentivirus-containing supernatants generated in HEK293T supplemented with 1 μg/ml polybrene (Sigma, H9268) for 8 h. Cells were then selected

**Fig. 6 | Mettl3 modulates the Hnf4a-centered regulatory network by control-ling *Hnf4a* mRNA stability. a** Gene set enrichment analysis (GSEA) for Hnf4a pathways in RNA-sequencing data of Control (Ctrl) and *Mettl3* cKO (cKO) mouse liver tissues at 1 day, 1 week, and 4 weeks birth. **b** Dual-luciferase reporter assays showing the effects of Mettl3 on *Hnf4a* reporters with either WT or mutated m⁶A-modified sites (*n* = 3 for each group over 3 independent experiments). **c** RT-qPCR for mRNA levels of *Hnf4a* at different time points in Control and *Mettl3* cKO mouse liver tissues (*n* = 4 for each group over 3 independent experiments). **d** Western blot for Hnf4α from Control and *Mettl3* cKO mouse liver tissues at 1 week, 2 weeks, and 4 weeks after birth. **e** Densitometry analysis of western blot for Hnf4a from Control and *Mettl3* cKO mouse liver tissues at indicated time points (*n* = 6 for each group). **f** RT-qPCR analysis of *Hnf4a* mRNA levels in primary hepa-tocytes isolated from Control and *Mettl3* cKO mouse liver at different time points after 5 μM actinomycin D treatment (*n* = 4 for each group over 3 independent experiments). **g** RT-qPCR analysis of *HNF4A* mRNA levels in METTL3 knockdown HepG2 cells at different time points after 5 μM actinomycin D treatment (*n* = 4 for each group over 3 independent experiments). **h** RIP-qPCR detecting the binding of Igf2bp1 to *Hnf4a* in Control and *Mettl3* cKO mouse liver tissues at 2 and 4 weeks after birth (*n* = 4 for each group). **i** RT-qPCR analysis of *HNF4A* mRNA levels in IGF2BP1 knockdown HepG2 cells at different time points after 5 μM actinomycin D treatment (*n* = 4 for each group). **j** Schematic diagram showing two rescue strate-gies with AAV8-Hnf4a. **k** Representative H&E staining and BODIPY staining pho-tographs of 2-weeks-old liver sections from *Mettl3* cKO mice intravenous injected with AAV8-Ctrl and AAV8-Hnf4a at day two after birth (6 experiments were repe-ated independently with similar results). Scale bar = 20 μm. **l** Statistical histogram of BODIPY staining (*n* = 5 for AAV8-Ctrl group; *n* = 7 for AAV8-Hnf4a group). **m** Statistical histogram of Ki67 immunohistochemical staining (n = 3 for AAV8-Ctrl group; *n* = 4 for AAV8-Hnf4a group). **n** Survival curves of *Mettl3* cKO mice intra-venous injected with AAV8-Ctrl and AAV8-Hnf4a (*n* = 8 for AAV8-Ctrl group; *n* = 9 for AAV8-Hnf4a group). Data in **b**, **c**, **e**–**i**, and **l**–**m** were shown as mean ± SEM with the indicated significance (*$P < 0.05$, **$P < 0.01$, ***$P < 0.001$; two-tailed student's *t*-test). Data in **n** were analyzed by Log-rank (Mantel-Cox) test with the indicated significance (***$P < 0.001$). Source data are provided as a Source Data file.

with 1.5 μg/ml puromycin (Thermo Scientific, A1113803) for two con-secutive days. TRC lentiviral vectors encoding shRNAs against human *METTL3* and human *IGF2BP1* were listed in Supplementary Table 1.

## Western Blot

Tissue samples were lysed with RIPA buffer (50 mM Tris-HCl (PH 7.4), 150 mM NaCl, 0.1% SDS, 1% Triton X-100, 1% sodium deoxycholate, and 2 mM EDTA (PH 8.0)) containing protease inhibitor (Roche, 04693132001) and phosphatase inhibitor (Roche, 04906837001). For cells, samples were counted, washed twice with ice-cold PBS, and lysed the same as tissues. Then the lysates were separated with SDS-PAGE gels and transferred to nitrocellulose membranes. Membranes were blocked with TBS containing 5% (v/w) non-fat milk and 0.1% Tween-20 (Sigma-Aldrich, P1379) and incubated with primary and secondary antibodies sequentially. Protein bands were detected using Immobilon ECL Ultra Western HRP Substrate (Millipore, WBULS500) according to the manufacturer's instructions. Gapdh or β-actin was used as the loading control. Uncropped blots were supplied in the Source Data file. Antibodies used for western blot were listed in Supplementary Table 2.

## RNA extraction and RT-qPCR

Total RNAs were extracted from tissues or cells using TRIzol (Ambion) according to the manufacturer's instructions and quantified by UV spectrophotometry. Reverse transcription was conducted using Pri-meScript™ RT Reagent Kit with gDNA Eraser (Perfect Real Time) (Takara, RR047B). RT-qPCR was then performed in triplicates on Light Cycler 480 II (Roche) using ChamQ Universal SYBR qPCR Master Mix (Vazyme, Q711-03). *Gapdh* was used as the internal control. The pri-mers used for RT-qPCR were listed in Supplementary Table 1.

## H&E, Masson's trichrome, and Immunohistochemistry staining

Liver samples were fixed in 4% paraformaldehyde and embedded with paraffin. Samples were sliced into 8 μm in thickness and then subjected to hematoxylin and eosin (H&E) staining or Masson's trichrome staining. For immunohistochemistry, sections were dewaxed, rehy-drated, and then incubated in EDTA antigen retrieval buffer (ZSGB-BIO, ZLI-9072) for 5 min at 100 °C. Slices were then incubated with 3% H₂O₂ for 10 min, washed three times with PBS containing 0.02% Tri-ton™ X-100 (Sigma-Aldrich, T8787), followed by incubation with pri-mary antibodies overnight at 4 °C. Horseradish peroxidase-conjugated antibody was used as the secondary antibody and incubated at 37 °C for 1 h. The color was developed by incubation with Dako Real™ kit (Dako, K5007). Sections were counterstained with hematoxylin (Baso, BA4041) and checked under the microscope (Nikon). Primary anti-bodies used in the immunohistochemistry staining were listed in Supplementary Table 2. Quantification for Masson's trichrome staining positive area and αSMA immunohistochemistry staining positive area

was conducted with 5-8 random fields (10*) each mouse using ImageJ software (version 1.8.0).

## TUNEL assay

Liver tissues were fixed, embedded with OCT compound (Servicebio, G6059), and sliced into 8 μm in thickness, then permeated with PBS containing 0.25% Triton™ X-100 (Sigma-Aldrich, T8787) and stained with In Situ Cell Death Detection Kit (Roche, 11684795910) following the manufacturer's instructions. Hoechst 33342 (Beyotime, C1022) was used to counterstain nuclei. Sections were visualized under the con-focal microscope (ZEISS, LSM 880), and images were analyzed by ZEN 2012 software. Quantification for TUNEL⁺ cells/Hoechst 33342⁺ cells ratio and average nuclear diameter was conducted using ImageJ soft-ware (version 1.8.0).

## PI staining

PI staining was performed on primary hepatocytes. The culture med-ium was removed, and cells were washed twice with pre-warmed PBS. Then cells were stained with propidium iodide solution (BD, 556547) for 15 min at 37 °C. Hoechst 33342 (Beyotime, C1022) was used to counterstain nuclei. Cells were detected under the microscope (Zeiss, Axio Observer Z1), and images were analyzed by ZEN 2012 software.

## Oil Red O staining

Liver tissues were fixed, embedded with OCT compound, and sliced into 8 μm in thickness. Oil Red O (Sigma-Aldrich, O0625) powder was dissolved in isopropanol at 0.7 g/100 mL concentration and then diluted with water at the volume ratio of 3:2 to get the Oil Red O solution. Pre-warmed tissue sections were washed twice with PBS and stained with Oil Red O solution for 20 min, then washed with 60% isopropanol for 10 s three times, followed by counterstaining with hematoxylin (Baso, BA4041), and then viewed under the microscope (Nikon). Oil Red O staining positive area was quantified with 5-8 ran-dom fields (10*) for each mouse using ImageJ software (version 1.8.0).

## BODIPY staining

BODIPY staining was performed on frozen liver sections and primary hepatocytes. For frozen liver tissue sections, liver tissues were fixed, embedded with OCT compound, sliced into 8 μm in thickness, and then stained with BODIPY (Invitrogen, D3922) at 7.6 μM for 30 min. Hoechst 33342 (Beyotime, C1022) was used to counterstain nuclei. Sections were mounted and scanned under the confocal microscope (ZEISS, LSM 880), and images were analyzed by ZEN 2012 software. For primary hepatocytes, the culture medium was removed, and the cells were washed twice with pre-warmed PBS. Then the cells were stained the same as liver tissues and checked under the microscope (Zeiss, Axio Observer Z1).

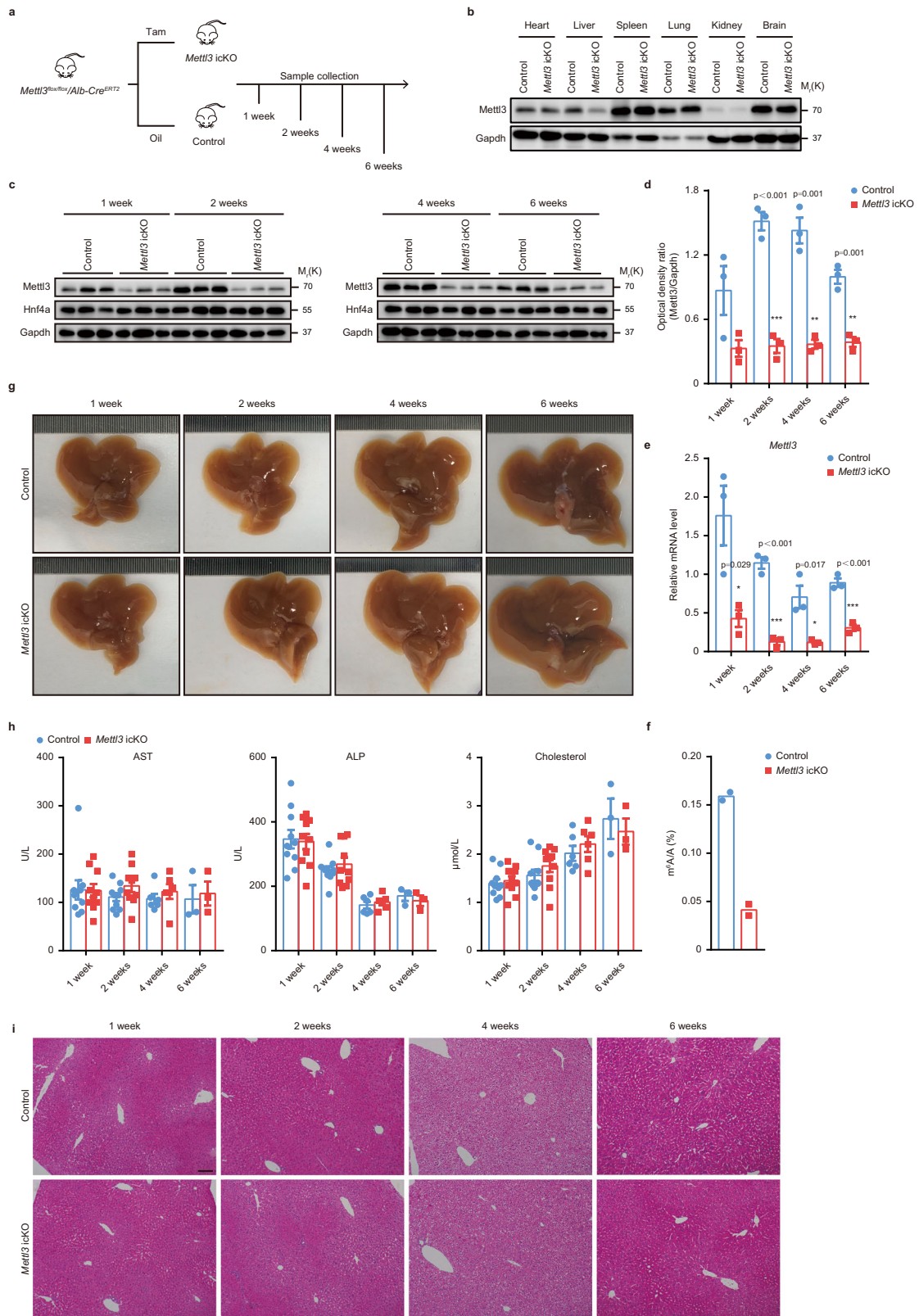

## mRNA stability assay

mRNA stability assay was performed on primary hepatocytes and HepG2 cell line. Primary hepatocytes were isolated from Control and hepatic *Mettl3* cKO mice. Cells were cultured overnight and then treated with 5 μM actinomycin D (Sigma-Aldrich, A1410) for 0 h, 1 h, 2 h, 4 h, and 6 h. HepG2 cells were cultured and treated with 5 μM actinomycin D for 0 h, 40 min, 2 h, and 4 h. For RT-qPCR analysis, total

RNAs were isolated and subjected to reverse transcription, and the mRNA levels of genes of interest were detected by RT-qPCR. For RNA stability RNA-sequencing, total RNAs were sent for RNA-sequencing.

## RNA-sequencing and m⁶A-RIP sequencing

RNA-sequencing and m⁶A-RIP sequencing for liver tissues were conducted by Guangzhou Epibiotek Co., Ltd. Briefly, for liver tissue RNA-

**Fig. 7 | Mettl3 is dispensable for homeostasis of adult liver. a** Schematic diagram of tamoxifen-induced liver-specific *Mettl3* knockout (*Mettl3* icKO) mouse generation. 4 weeks old *Mettl3^flox/flox^/Alb-Cre^ERT2^* mice were treated with tamoxifen (Tam) or olive oil (Oil) for 5 consecutive days and then labeled with 0 week. **b** Western blot for Mettl3 in indicated tissues from Control and *Mettl3* icKO mice at 1 week after tamoxifen treatment (3 experiments were repeated independently with similar results). **c** Western blot for Mettl3 and Hnf4a in Control and *Mettl3* icKO mouse liver tissues at different time points after tamoxifen treatment (3 experiments were repeated independently with similar results). **d** Densitometry analysis of western blot for Mettl3 from Control and *Mettl3* icKO mouse liver tissues at different time points after tamoxifen treatment (*n* = 3 for each group). **e** RT-qPCR for *Mettl3* expression in liver tissues at different time points after tamoxifen treatment (*n* = 3 for each group over 3 independent experiments). **f** LC–MS/MS analysis of m⁶A/A

ration in Control and *Mettl3* icKO mouse livers 2 weeks after tamoxifen treatment (*n* = 2 for each group). **g** Representative gross appearance of livers from Control and *Mettl3* icKO mice at 1 week, 2 weeks, 4 weeks, and 6 weeks after tamoxifen treatment (3 experiments were repeated independently with similar results). **h** Serum levels of AST, ALP, and Cholesterol of Control and *Mettl3* icKO mice at different time points after tamoxifen treatment (*n* = 3 for 6 weeks group; *n* = 6 for 4 weeks group; *n* = 10 for other groups). **i** Representative H&E staining photographs of Control and *Mettl3* icKO mouse liver sections at indicated time points (3 experiments were repeated independently with similar results). Scar bar = 100 μm. Data in **d**–**f** and **h** were shown as mean ± SEM with the indicated significance (*P < 0.05, **P < 0.01, ***P < 0.001; two-tailed student's t-test). Source data are provided as a Source Data file.

sequencing, total RNA was isolated from Control and *Mettl3* cKO mouse livers at day 1, 1 week, 2 weeks, 4 weeks, and 8 weeks postnatally (2 individuals for each time point). Sequence reads were aligned to the mouse genome version mm10 with HISAT2.1.0[60], and differentially expressed genes (DEGs) were calculated by DESeq2[61] under the following criteria: |log2FC| > 1 and P value < 0.05. Gene set enrichment analysis (GSEA) was conducted using the GSEA software (https://www.broadinstitute.org/gsea/)[62]. For mouse primary hepatocyte RNA stability RNA-sequencing, total RNA was isolated from Control and *Mettl3* cKO primary hepatocyte at 0 h, 2 h, and 6 h treatment with 5 μM actinomycin D and sequenced by BerryGenomics Company (http://www.berrygenomics.com/, Beijing, China). Reads were mapped to the mouse mm10 genome with STAR v2.5.3a[63], and the uniquely mapped reads with q score no less than 250 were kept. Gene counting was performed by featureCounts v2.0.1 with exon features documented in the Gencode mouse annotation gtf file[64]. Differentially expressed genes between WT and KO at different time points were identified by edgeR package[65]. For m⁶A-RIP sequencing, total RNA was isolated from WT mouse livers at day 1, 1 week, 2 weeks, 4 weeks, and 8 weeks postnatally (2 individuals for each time point) using TRIZOL reagent and fragmented. m⁶A-modified RNA was enriched by m⁶A antibody and rRNA was removed. The library was prepared by smart-seq method and sequenced. Sequence reads were aligned to the mouse genome version mm10 with HISAT2.1.0. Differential m⁶A-modified peaks between RIP and input samples were identified using exomePeak[66]. The longest isoform was retained if a gene had more than one isoform. Motif search was conducted with HOMER[67].

## LC-MS/MS for m⁶A detection and quantification

Bulk mRNA m⁶A modification quantification through LC-MS/MS was conducted by Wuhan Metware Co., Ltd. Briefly, 1 μg purified mRNA was sufficiently digested to nucleosides with S1 nuclease, phosphodiesterase, and alkaline phosphatase in 37 °C, then extracted by chloroform to get prepared solution samples. The samples were analyzed using a UPLC-ESI-MS/MS system (UPLC, ExionLC™ AD, https://sciex.com.cn/; MS, Applied Biosystems 6500 Triple Quadrupole, https://sciex.com.cn/). The effluent was alternatively connected to an ESI-triple quadrupole-linear ion trap (QTRAP)-MS. Linear ion trap (LIT) and triple quadrupole (QQQ) scans were acquired on a triple quadrupole-linear ion trap mass spectrometer (QTRAP) equipped with an ESI Turbo Ion-Spray interface, then operated in a positive ion mode and controlled by Analyst 1.6.3 software (Sciex). RNA modifications were analyzed using scheduled multiple reaction monitoring (MRM). Data acquisitions were performed using Analyst 1.6.3 software (Sciex). RNA modification contents were detected by MetWare (http://www.metware.cn/) based on the AB Sciex QTRAP 6500 LC-MS/MS platform.

## m⁶A-RIP-qPCR

m⁶A-RIP-qRCR was performed according to previous reports[68,69]. Briefly, total RNAs were extracted from 2 weeks or 4 weeks old Control and *Mettl3* cKO mice. Poly(A) mRNAs were separated using an mRNA

purification kit (Sigma-Aldrich, GenElute™ mRNA Miniprep Kit MRN10). Two rounds of purification process were conducted in each experiment to sufficiently remove rRNA contamination according to manufacturer's instruction. 5 μg mRNAs were fragmented into 200-300 nt segments by incubation at 94 °C for 30 s in fragmentation buffer (10 mM ZnCl₂, 10 mM Tris-HCl (PH 7.0)), then stopped with 50 mM EDTA, and purified by ethanol precipitation. Fragmented mRNAs and 2 μg anti-m⁶A antibody (Synaptic Systems, 202003) or mouse IgG (Beyotime, A7028) were added in 600 μL RIP buffer (150 mM NaCl, 0.1% Igepal CA-630, 10 mM Tris-HCl (PH 7.4)) and incubated at 4 °C for 2 h. Then 15 μL Dynabeads® Protein A beads (Thermo, 100-02D) and 15 μL Dynabeads® Protein G beads (Thermo, 100-04D) were added to the mixture and incubated at 4 °C for another 2 h. Beads were washed with RIP buffer 5 times. RNase inhibitor (Promega, N2611) was added throughout the entire process. Fragmented mRNAs were eluted by 100 μL 0.3 μg/μL Proteinase K (Thermo Scientific, AM2546) at 55 °C for 1 h, and followed by phenol-chloroform extraction and ethanol precipitation purification procedure. The precipitated mRNAs were reverse transcribed, and enrichment was determined by RT-qPCR as mentioned above. RT-qPCR primers for m⁶A positive regions were marked as "positive-m⁶A-RT" (e.g., primers for two separate m⁶A positive regions in *Ppara* were listed as *Ppara*-positive-m⁶A-1-RT and *Ppara*-positive-m⁶A-2-RT), and primers for m⁶A negative regions were marked as "negative-m⁶A-RT" (e.g., the primer for m⁶A negative regions in *Ppara* were listed as *Ppara*-negative-m⁶A-RT). Primers used for m⁶A-RIP-qPCR were listed in Supplementary Table 1.

## RNA immunoprecipitation (RIP)-qPCR

RIP was performed according to previously published protocols with modifications[69,70]. Briefly, 200 mg liver tissues were homogenized with 600 μL homogenizer buffer (100 mM KCl, 5 mM MgCl₂, 10 mM HEPES, pH 7.0, 0.5% Nonidet P-40, 1 mM DTT, and 100 U/mL RNase inhibitor (Promega, N2611), then incubated on ice for 5 min, followed by centrifuging to obtain supernatant. 2 μg Igf2bp1 antibody was added to 15 μL Dynabeads® Protein A beads (Thermo, 100-02D) and 15 μL Dynabeads® Protein G beads (Thermo, 100-04D), then rotated at 4 °C for 2 h. Antibody-beads slurry was then incubated with homogenized supernatant and rotated at 4 °C for another 2 h. Beads were washed 5 times with Washing Buffer (50 mM Tris-HCl (pH 7.4), 150 mM NaCl, 1 mM MgCl₂, and 0.05% NP40), then extracted RNA using Trizol for RNA extraction and further analysis for RT-qPCR.

## Dual-luciferase reporter assays

To construct the Mettl3 overexpression vector, the full-length coding sequences of mouse Mettl3 were amplified by PCR with Phanta Max DNA Polymerase (Vazyme, P505) using *Mettl3*-PKD-F and *Mettl3*-PKD-R primer, then cloned into a lentiviral vector PKD-EF1 using ClonExpress II One Step Cloning kit (Vazyme, C112). Mettl3-D395A-W398A catalytic mutant (DPPW/APPA) vector was constructed using the site-directed mutagenesis method (using PCR-based method with Mettl3

overexpression vector as template and *Mettl3*-AWWA-F and *Mettl3*-AWWA-R as primers). PKD-EF1 vector expressing EGFP served as a control in transfection. DNA fragments of *Hnf1a* and *Hnf4a* containing the WT m⁶A motifs and mutant motifs (Supplementary Fig. 6c) were synthesized by Shanghai Generay Biotech Co., LTD., and then subcloned into pMIR-REPORT firefly luciferase reporter vector (Ambion, AM5795) between Mlu I and Sac I sites. All sequences were confirmed by Sanger sequencing. 50 ng Mettl3 vectors (Mettl3 overexpressing vector or control), 40 ng firefly luciferase reporter vectors with WT or mutated fragments, and 10 ng pRL Renilla Luciferase Control Reporter Vector (Promega, E2231) were co-transfected into HEK293T cells in triplicates in 96-well plates. Fluc and Rluc activities were measured 24 h later with the Dual-Luciferase Reporter Assay System (Promega, E1910) according to the manufacturer's instructions. The relative luciferase activity was calculated through dividing Fluc activity by individual Rluc activity and then normalizing to control of each assay. The sequences of PCR primers were listed in Supplementary Table 1.

### Reporting summary
Further information on research design is available in the Nature Research Reporting Summary linked to this article.

## Data availability
Source data are provided with this paper. RNA-sequencing and m⁶A-RIP sequencing raw data and processed expression matrix are uploaded to GEO DataSets under accession code GSE197564. The sequencing reads were mapped to the mouse mm10 genome. All other data analyzed or generated in this study are provided along with the article. Source data are provided with this paper.

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

## Acknowledgements

We show our full respect and gratitude to all members of Qi Zhang and Yan Xu's lab for their discussion and technical assistance. This work was supported by the National Key Research and Development Program of China (No. 2017YFA0106100, X.Y.), National Natural Science Foundation of China (No. 81970537, X.Y.; No. 31601184, X.Y.; No. 81870449, Z.Q.; No. 82170674, Z.Q.), Guangdong Basic and Applied Basic Research Foundation (No. 2020A1515011385, X.Y.), Guangzhou Science and Technology Plan Project (No. 202206010072, Z.Q.), the Fundamental Research Funds for the Central Universities, Sun Yat-sen University (No. 22ykqb02, X.Y.), and Guangdong Science and Technology Program (2020B1212060019).

## Author contributions

Q.Z. and Y.X. conceived the idea, supervised the study, analyzed data, and acquired funding. Y.X. drafted the manuscript with the help of Z.Z. and X.K. Q.Z. critically revised the manuscript. Z.Z. and X.K. contributed to the experimental design and conducted most experiments with the help of L.P., C.L., X.L., J.C., S.D., Y.L., Q.L., Y.S., and S.Y. The manuscript was read and approved by all authors.

## Competing interests

The authors declare no competing interests.
