## [Peer Review File · Nature Communications]

REVIEWER COMMENTS

Reviewer #1 (Remarks to the Author):

Previous studies show that N6-methyladenosine (m6A) modification on mRNAs plays a critical role in development of several mouse tissues, and is dynamically regulated during mammalian liver development. In the current manuscript, the authors investigated the role of m6A on mRNA during mouse liver development. They illustrate that the catalytic subunit of m6A writer complex Mettl3 is essential for liver development using a liver-specific Mettl3 knockout mouse model. They show that m6A is enriched on a panel of liver developmental genes to stabilize the mRNAs and promote the protein expression to support liver development. Furthermore, the authors found that Mettl3 is required only during liver development, but not to maintain homeostasis of adult liver using the inducible liver-specific Mettl3 knockout mouse. Together, this manuscript shed light on a critical role of m6A during mouse liver development.

The authors demonstrate strong evidence to support an important role of m6A on mRNA during liver development. Their results are consistent with other developmental studies suggesting an essential function of the m6A writer complex during various development systems including hematopoietic stem cells and embryonic stem cells. Despite the authors carefully illustrating the role of m6A on the phenotypes of liver development, the current manuscript lacks the mechanisms to understand the role of m6A, which limits the scope of the study.

Based on these issues, the manuscript needs additional work focused on the mechanistic aspect of m6A function in order to be suitable for publication. My specific concerns are enumerated below.

Major

1. The authors demonstrated the important role of Mettl3 during mouse liver development. They revealed that the major Mettl3 target related to liver development is Hnf4 α by stabilizing the mRNA. However, their study lacks a mechanism to explain how m6A on Hnf4 α regulates its expression. Without such mechanistic insights, the current manuscript is limited as an observational study.

This is particularly important since the major effect of m6A is to destabilize mRNAs via the YTHDF proteins. The effect of m6A on Hnf4 α is likely indirect. Numerous examples of m6A indirectly affecting an mRNA via regulating some other transcript have been described in the literature. Something similar is

likely operational here. However, some experiments that provide insight, preferably a clear mechanism, would strengthen this manuscript.

2. Related to Figure 5 and 5S. The authors quantify bulk m6A levels by dot blotting and m6A distribution by m6A-seq during development. m6A in the dot blot and Mettl3 expression peak at 2-4 weeks postnatal. Dot blotting is unacceptable since the m6A antibody also detects m6Am, another modification, and m6A in rRNA, which hasn't been shown to be excluded.

3. The number of m6A peaks consistently decreases after birth by m6A-seq. However, extrapolations of regulatory effects from m6A-Seq are not reliable and cannot be used since this method is inherently noisy with variable peak heights between biological replicates (see A. McIntyre and C. Mason, Scientific Reports, 2020). So more quantitative measurements, preferably on individual transcripts, are needed. Regardless, one possible explanation to this observation is that m6A is more enriched at each site after 3 weeks compared to day 1, although there are smaller number of m6A peaks at 3 weeks. The authors should clarify this point with additional analysis.

Minor concerns

1. Also important to note that the proper nomenclature of N6-methyladenosine is “m6A” (“6” as superscripted).

Part 5: Method

1) In the description of m6A dot-blotting, the authors isolated total RNA, normalized the concentration, denatured, and applied to dot blotting using the anti-m6A antibody. rRNA is highly abundant and highly methylated RNA in total RNA samples. Indeed, m6A signals from rRNA is a major obstacle in the field to accurately measure m6A levels in mRNA. If isolating mRNA from total RNA samples, at least two rounds of mRNA isolation is the standard to sufficiently remove rRNA contamination. If mRNA isolation step was not included in this method, I highly doubt that the authors could measure m6A in mRNA. Please clarify and revise the method.

2) In the section of m6A-RIP sequencing, the authors describe that total RNA was isolated and fragmented, then enrichment for m6A-modified mRNA. How did they enrich mRNA after fragmenting total RNA? Please clarify.

Reviewer #2 (Remarks to the Author):

m6A modification is involved in many aspects of mammalian development and disease (PMID: 33611339). Many mRNAs are altered because of knockout of Mettl3 in hepatocytes (Mettl3 cKO). The authors focused on HNF4A in this study, since this factor is reduced in the liver-specific Mettl3-null mice and is known to control the expression of many critical genes involved in liver development and liver function. Both mouse lines have hepatic steatosis, fibrosis, and liver damage. The phenotype of the Mettl3 cKO mice appear to phenocopy the hepatocyte-specific Hnf4a-null mice but this was not directly demonstrated. Surprisingly, in the adult hepatocyte-specific Mettl3-null mice (Mettl3 icKO), loss of Mettl3 did not appear to result in a liver phenotype. While this study does show the involvement of Mettl3 in early liver development in part through control of HNF4A expression, mechanistic studies are lacking including how Mettl3 is induced and then suppressed after birth and how it controls HNF4A only during the developmental stage and not in the adult hepatocyte.

Major issues:

1. Figure 1B: Mettl3 is decreased significantly at day 1 even though the albumin promoter (and Alb-Cre are not expressed until 1-2 weeks postnatally. Explain this discrepancy.
2. Figure 1C: What is the postnatal day that these samples were collected?
3. Figure 1C shows only a very faint band in the Mettl3 cKO while SFig. 1e shows an incomplete knockout of Mettl3 in liver. What is the explanation for this difference? Hepatocytes should be purified and analyzed to establish that the Mettl3 gene is knocked out in hepatocytes and the extent of knockout.
4. Page 10, line 210: "...most Hnf4 α target genes involved in liver development, such as Apoa2, Apoc3, Cyp8b1, and Mttp.....". There is no evidence that Apoa2, Apoc3, Cyp8b1, and Mttp are involved in liver development. Which are the HNF4A target genes that are responsible for the liver phenotype and lethality phenotype? This should be determined by rescue experiments.

5. As the authors note, the liver phenotype and the lethality are much more severe than that found with the hepatocyte-specific Hnf4a-null mice, also made with Alb-Cre. This indicates that there are other factors involved that are modified by Mettl3 that have a role in liver development. RNA-seq should be carried out on hepatocyte-specific Hnf4a-null mice and Mettl3 cKO mice.

6. The evidence (Figure 6H) that m6A modification of mRNA increases its degradation is not convincing. The decrease in mRNA is modest and could be the result of actinomycin-induced liver toxicity. An RNA-seq could be carried out to see if the effect is non-specific. It is just as likely that another mRNA or a miRNA that is modified by Mettl3 alters levels of Hnf4a mRNA. Or the possibility cannot be ruled out that the modification alters Hnf4a pre-RNA processing, transport of the Hnf4a mRNA to the cytoplasm of Hnf4a mRNA translation.

7. Fig. S6A, Supplementary Table 3: The RNA-seq of the Control and Mettl3 cKO livers should be compared with hepatocyte-specific Hnf4a-null livers.

8. There is a large variability in the HNF4A protein in Figure 6D. These blots should be quantified.

9. Figure 7: Hnf4a mRNA and its target gene mRNAs should be measured in this model.

10. m6A modification levels should be measured in the adult Mettl3^{flox/flox}/Alb-CreERT2 (Mettl3 icKO) mice with and without tamoxifen.

11. Does knockout of Mettl3 in adult mice alter liver regeneration or response to chemically-induced liver injury?

12. Page 14, line 299: "In fact, most of the abnormalities we found in Mettl3 cKO mice (including lipid deposition in liver, increased bile acids in serum, liver injury and lethality in young adults) phenocopied that of hepatic Hnf4 α knockout mice." Provide a citation for this statement.

Minor issues:

13. Nomenclature: There are no symbols used in gene, mRNA, and protein nomenclature. Hnf4 α , Hnf1 α , Ppara, and Cebpa should be Hnf4a, Hnf1a, Ppara, and Cebpa

14. Figure 7F: The numbers in this table should be reduced to three significant figures.

15. METTL3 regulates heterochromatin in mouse embryonic stem cells (PMID: 33505026). This paper should be discussed as a possible mechanism that gives rise to the phenotype in the hepatocyte-specific Mettl3-null mice.

16. It is unconventional and distracting for figures to be re-cited in the Discussion, other than a model figure.

17. Figure 8 could be deleted or put in Supplemental figures.

Remarks to the authors:

Reviewer #1:

Previous studies show that N6-methyladenosine (m⁶A) modification on mRNAs plays a critical role in development of several mouse tissues, and is dynamically regulated during mammalian liver development. In the current manuscript, the authors investigated the role of m⁶A on mRNA during mouse liver development. They illustrate that the catalytic subunit of m⁶A writer complex Mettl3 is essential for liver development using a liver-specific Mettl3 knockout mouse model. They show that m⁶A is enriched on a panel of liver developmental genes to stabilize the mRNAs and promote the protein expression to support liver development. Furthermore, the authors found that Mettl3 is required only during liver development, but not to maintain homeostasis of adult liver using the inducible liver-specific Mettl3 knockout mouse. Together, this manuscript shed light on a critical role of m⁶A during mouse liver development.

The authors demonstrate strong evidence to support an important role of m⁶A on mRNA during liver development. Their results are consistent with other developmental studies suggesting an essential function of the m⁶A writer complex during various development systems including hematopoietic stem cells and embryonic stem cells. Despite the authors carefully illustrating the role of m⁶A on the phenotypes of liver development, the current manuscript lacks the mechanisms to understand the role of m⁶A, which limits the scope of the study.

Response: We thank the reviewer for evaluating our paper carefully and giving us positive comments and valuable suggestions. We agree with the reviewer and conducted more experiments to make our conclusion more compelling now. We hope that the reviewers will be satisfied with the revised version of our manuscript.

Major

1. The authors demonstrated the important role of Mettl3 during mouse liver development. They revealed that the major Mettl3 target related to liver development is Hnf4 α by stabilizing the mRNA. However, their study lacks a mechanism to

explain how m⁶A on Hnf4a regulates its expression. Without such mechanistic insights, the current manuscript is limited as an observational study. This is particularly important since the major effect of m⁶A is to destabilize mRNAs via the YTHDF proteins. The effect of m⁶A on Hnf4a is likely indirect. Numerous examples of m⁶A indirectly affecting an mRNA via regulating some other transcripts have been described in the literature. Something similar is likely operational here. However, some experiments that provide insight, preferably a clear mechanism, would strengthen this manuscript.

Response: We are grateful for the reviewer pointing out the lack of a detailed mechanism of how m⁶A on Hnf4a regulates its expression in our study. We agree with the reviewer and have conducted more experiments to demonstrate it.

m⁶A modification controls RNA fate mainly through “reader” proteins specifically recognizing and binding to m⁶A-containing transcripts. Two prominent protein families are responsible for recognizing m⁶A modifications: YTH521-B homology (YTH) domain-containing protein family and insulin-like growth factor-2 (IGF2) family. Detailed function of each “reader” was summarized below (**Attached Table 1**).

Name	Localization	Function	References
YTHDF1	Cytosol	Enhance mRNA translation	Cell . 2015 Jun 4; 161(6): 1388-99
YTHDF2	Cytosol	Promote mRNA degradation	Cell . 2015 Jun 4; 161(6):1388-99 Nat Commun . 2016 Aug 25;7:12626 Nature . 2014 Jan 2;505(7481):117-20
YTHDF3	Cytosol	Enhance translation and degradation by interacting with YTHDF1 and YTHDF2	Cell Res . 2017 Mar;27(3):315-328 Cell Res . 2017 Mar;27(3):444-447
YTHDC1	Nucleus	mRNA splicing/nuclear export	Mol Cell . 2016 Feb 18; 61(4):507-519 eLife . 2017 Oct 6;6:e31311
YTHDC2	Nucleus and Cytosol	Enhance the translation efficiency and decrease mRNA abundance	Cell Res .2017 Sep;27(9):1115-1127

IGF2BP1, IGF2BP2, IGF2BP3	Nucleus and Cytosol	Enhance stability and translation	mRNA and	Nat Cell Biol. 2018 Mar;20(3):285-295
------------------------	--	-------------	--

Attached Table 1. Summary of the function of m⁶A “readers”

Our work found that decreased m⁶A on *Hnf4a* mRNA led to Hnf4a down-regulation on both mRNA and protein levels (new **Fig. 6c** and **6d**). Among m⁶A “readers” identified so far, IGF2BPs promote the stability and storage of their target transcripts (*Nat Cell Biol.* 2018 Mar;20(3):285-295). Thus, we checked previous publications and found that deletion of IGF2BP1 leads to destabilization of *Hnf4a* mRNA in HepG2 cells, while interfering with other two members did not affect *Hnf4a* mRNA degradation (*Nat Cell Biol.* 2018 Mar;20(3):285-295) (new **Fig. S7j**), indicating that IGF2BP1 may directly recognize m⁶A on *Hnf4a* mRNA and maintain its mRNA levels in the context of liver.

We conducted further experiments to verify our hypothesis. Firstly, we confirmed the direct binding of Igf2bp1 to *Hnf4a* transcripts in both 2 weeks and 4 weeks mouse liver tissues by RNA immunoprecipitation (RIP)-qPCR (new **Fig. 6h**). In addition, we observed a decreased enrichment of Igf2bp1 in *Mettl3* cKO individuals (please see new **Fig. 6h**). Furthermore, knocking down IGF2BP1 in HepG2 cells led to the destabilization of *Hnf4a* mRNA, similar to *Mettl3* knockout in primary hepatocytes and *METTL3* knockdown in HepG2 cells (please see new **Fig. 6f, 6g, and 6i**).

We also conducted RNA-sequencing to see global changes of mRNA half-life in *Mettl3* cKO hepatocytes compared to the wild-type control. As the reviewer mentioned, mRNA of most genes became more stable in *Mettl3* cKO hepatocytes, especially those with m⁶A modifications (please see new **Supplementary Fig. 7g, 7h, and Supplementary Table 4**), consistent with previous reports (*Nature.* 2014 Jan 2;505(7481):117-20; *Science.* 2015 Feb 27;347(6225):1002-6). However, we observed an apparent decrease of *Hnf4a* mRNA half-life when *Mettl3* was blocked (please see new **Supplementary Fig. 7i**). Along with the results of m⁶A-RIP-sequencing, m⁶A-RIP-qPCR, and dual-luciferase reporter assays (**Fig. 5d**,

5e, and Fig. 6b), all of which showed authentic m⁶A deposition on *Hnf4a* transcripts in the liver, we concluded that the effect of m⁶A on *Hnf4a* is direct and inhibition of Mettl3-mediated m⁶A in the liver reduced mRNA stability of *Hnf4a* through an IGF2BP1-dependent way. We have added the corresponding data into new Fig. 6h, 6i and Supplementary Fig. 7g-7i. We also modified the text accordingly (please see Page 11 line 229 to Page 13 line 265).

To further exclude other possibilities of m⁶A controlling *Hnf4a* expression (such as mRNA transportation and splicing), we isolated nuclear and cytoplasm fractions of liver tissues and primary hepatocytes and found that *Hnf4a* mRNA levels in both fractions were comparable (please see new Supplementary Fig. 7b-7d). We also re-analyzed RNA-sequencing of control and *Mettl3* cKO liver tissues at different developmental time points, and no difference of RNA splicing was observed (data not shown). These results further support our conclusion that decreased *Hnf4a* expression after *Mettl3* knockout in the liver is mainly because of the destabilization of *Hnf4a* transcripts.

2. Related to Figure 5 and 5S. The authors quantify bulk m⁶A levels by dot blotting and m⁶A distribution by m⁶A-seq during development. m⁶A in the dot blot and *Mettl3* expression peak at 2-4 weeks postnatal. Dot blotting is unacceptable since the m⁶A antibody also detects m⁶Am, another modification, and m⁶A in rRNA, which hasn't been shown to be excluded.

Response: We apologize for the imprecise approach we used in the original manuscript. We thank the reviewer for pointing this out. We extracted mRNA by a two-round purification process, which can effectively exclude rRNA in samples according to the manufacturer's instructions. To exclude the interference of m⁶Am, we subjected the purified mRNA samples to LC-MS/MS, which could distinguish m⁶A and m⁶Am and precisely quantify the level of m⁶A in each sample. The results demonstrated significant reduction of mRNA m⁶A level in *Mettl3* cKO and *Mettl3* icKO mouse livers compared to that of control (please see new Fig. 7f and Supplementary Fig. 11). LC-MS/MS results also showed increased m⁶A level after

birth, peaked at 2 weeks, and decreased then (please see new **Supplementary Fig. 5a**). Since the inaccuracy of dot blotting to reflect the bulk m⁶A levels as the reviewer said, we eliminated the results of dot blotting.

3. The number of m⁶A peaks consistently decreases after birth by m⁶A-seq. However, extrapolations of regulatory effects from m⁶A-Seq are not reliable and cannot be used since this method is inherently noisy with variable peak heights between biological replicates (see A. McIntyre and C. Mason, *Scientific Reports*, 2020). So more quantitative measurements, preferably on individual transcripts, are needed. Regardless, one possible explanation to this observation is that m⁶A is more enriched at each site after 3 weeks compared to day 1, although there are smaller number of m⁶A peaks at 3 weeks. The authors should clarify this point with additional analysis.

Response: Many thanks for the reviewer's constructive suggestions. We agree with the reviewer that the peak number cannot reflect the total level of m⁶A modification. We apologize for the errors we made in the original manuscript. We also agree with the reviewer that m⁶A-RIP-seq has many shortcomings (such as inherently noise, interference of m⁶Am, and low resolution), and more and more newly developed technologies, like miCLIP-seq, m⁶A-LAIC-seq, and m⁶A-REF-seq are emerging (*Nat Methods*. 2015 Aug;12(8):767-72, *Nat Methods*. 2016 Aug;13(8):692-8, *Sci Adv*. 2019 Jul 3;5(7):eaax0250). It is a pity that m⁶A-RIP-seq is still the most widely used technique to determine global distribution patterns of m⁶A modification so far. As the reviewer suggested, we did additional analyses to compare enrichment of m⁶A at different time points. The results showed that the global m⁶A peak enrichment increased after birth, peaked at 2 weeks, and started to decrease then, similar with protein level changes of Mettl3 (**Fig. S1A**) and bulk m⁶A levels (new **Supplementary Fig. 5a**) during mouse liver development. We have added these data into the new **Supplementary Fig. 5d** and modified the text accordingly (please see **Page 8, line 172-174**).

Minor concerns:

1. Also important to note that the proper nomenclature of N6-methyladenosine is “m⁶A” (“6” as superscripted).

Response: We apologize for the careless errors in the original manuscript. In the current version, we have amended it and carefully double-checked the entire manuscript accordingly.

Part 5: Method

1) In the description of m⁶A dot-blotting, the authors isolated total RNA, normalized the concentration, denatured, and applied to dot blotting using the anti-m⁶A antibody. rRNA is highly abundant and highly methylated RNA in total RNA samples. Indeed, m⁶A signals from rRNA is a major obstacle in the field to accurately measure m⁶A levels in mRNA. If isolating mRNA from total RNA samples, at least two rounds of mRNA isolation is the standard to sufficiently remove rRNA contamination. If mRNA isolation step was not included in this method, I highly doubt that the authors could measure m⁶A in mRNA. Please clarify and revise the method.

Response: We apologize for the imprecise description of the methods we used. Many thanks for the reviewer to point this out. We extracted mRNA by a standard two-round purification process to eliminate rRNA contamination and used purified mRNA for dot-blotting, m⁶A-RIP-sequencing, and m⁶A-RIP-qPCR experiments. We have now described adequately in the current Supplementary Materials and methods (please see **Page 12**). Considering the contamination of m⁶Am in dot-blotting as the reviewer mentioned above, we have replaced the dot-blotting results with LC-MS/MS results of purified mRNA (new **Fig. 7f**, **Supplementary Fig. 5a**, and **Supplementary Fig. 11**).

2) In the section of m⁶A-RIP sequencing, the authors describe that total RNA was isolated and fragmented, then enrichment for m⁶A-modified mRNA. How did they enrich mRNA after fragmenting total RNA? Please clarify.

Response: We are very sorry for the wrong description in Methods. When we conducted m⁶A-RIP-seq, we first enriched mRNA and then did fragmentation. We have now fixed it and described adequately in the methods section (please see

Supplementary Materials and methods page No. 11).

Reviewer #2

m⁶A modification is involved in many aspects of mammalian development and disease (PMID: 33611339). Many mRNAs are altered because of knockout of *Mettl3* in hepatocytes (*Mettl3* cKO). The authors focused on HNF4A in this study, since this factor is reduced in the liver-specific *Mettl3*-null mice and is known to control the expression of many critical genes involved in liver development and liver function. Both mouse lines have hepatic steatosis, fibrosis, and liver damage. The phenotype of the *Mettl3* cKO mice appear to phenocopy the hepatocyte-specific *Hnf4a*-null mice but this was not directly demonstrated. Surprisingly, in the adult hepatocyte-specific *Mettl3*-null mice (*Mettl3* icKO), loss of *Mettl3* did not appear to result in a liver phenotype. While this study does show the involvement of *Mettl3* in early liver development in part through control of HNF4A expression, mechanistic studies are lacking including how *Mettl3* is induced and then suppressed after birth and how it controls HNF4A only during the developmental stage and not in the adult hepatocyte.

Response: We sincerely thank the reviewer for evaluating our work carefully and giving us helpful suggestions.

The first concern of the reviewer is the lack of study on how *Mettl3* is regulated in liver development. We first did a prediction of transcription factor binding on the *Mettl3* promoter through TRANSFAC, EPD, and JASPAR. We found a group of targets that might bind to the *Mettl3* promoter, including two hepatic nuclear factors, *Foxa1* and *Foxa2*. Next, we searched the published papers and found that *Foxa1* and *Foxa2* in the liver increased with age after birth (*Genes Dev.* 2020 Aug 1;34(15-16):1039-1050.). We also did a western blot for *Foxa2* in liver tissues and observed a progressive increase with age (**Attached Fig. 1A**), which is not exactly the same as the expression pattern of *Mettl3* (**Supplementary Fig. 1a**). *Mettl3* might be regulated by different group of factors during different develop stages. Also, *Foxa* family members are usually thought to activate gene expression. These results

indicated that *Mettl3* dynamic expression might be potentially regulated by Foxa family members during the first two weeks of postnatal liver development, and other regulatory mechanisms could appear at later stages, and different mechanisms may account for dynamic changes of *Mettl3* during different developmental stages.

We also scanned for miRNAs potentially regulating *Mettl3* expression using miRDB, and Tarbase. From overlapped candidates, we figured out that several miRNAs (miR369, miR760, and miR877), which have been reported to play a role in the liver (*J Cancer*. 2021 Mar 19;12(10):3067-3076; *J Biochem Mol Toxicol*. 2018 Aug;32(8):e22167; *Int J Clin Exp Pathol*. 2015 Feb 1;8(2):1515-24.), showed high scores to bind to *Mettl3* transcripts, thus worth to be investigated in the liver development process. However, given the elaborate and intricate cellular and molecular context, it is hard to trace the upstream regulators of *Mettl3* in the dynamic development process *in vivo*. As the reviewer may also agree, we still lack effective *in vitro* models to study postnatal liver development, limiting us to verify these hypotheses effectively in the developmental context. Nevertheless, we believe it is valuable to figure out how *Mettl3* is dynamically regulated in the developmental process, and we will continue to work on this field.

Another point the reviewer raised is why *Mettl3* controlled *Hnf4a* only during the developmental stage and not in the adult hepatocytes. To see whether m⁶A modification in *Hnf4a* mRNA was changed in the *Mettl3* icKO adults, we conducted m⁶A-RIP-qPCR in control and *Mettl3* icKO livers collected 2 weeks after Tamoxifen induction and found that m⁶A modification on *Hnf4a* mRNA was also significantly reduced in icKO livers compared to that of Control (**Attached Fig. 1B**), similar to that in cKO mouse livers in early development (**Fig. 5e**). However, both mRNA and protein levels of *Hnf4a* (also *Hnf4a* targets) did not change in icKO livers at different time points after Tamoxifen treatment (**Fig. 7c-e**, and **Supplementary Fig. 8e-k**), indicating the dispensable role of m⁶A in regulating *Hnf4a* expression in the adult liver. Then we compared the expression pattern of known m⁶A “readers” between neonate and adult livers. The results showed that all “readers” were much lower in

adult livers than neonates (**Attached Fig. 1C**). We have now demonstrated that *Igf2bp1* was responsible for *Hnf4a* mRNA m⁶A recognition in postnatal liver development (new **Fig. 6h, 6i, and Supplementary Fig. 7j**) thus we verified *Igf2bp1* expression in livers from postnatal day 1 and 8-week-old mice with RT-qPCR and observed a similar decrease in adults as that in RNA-seq (**Attached Fig. 1D**). Therefore, among other possibilities, we think the differential expression of “readers” in the neonate and adult livers may at least partially explain why m⁶A controls *Hnf4a* only during the early development but not in the adult hepatocytes.

Attached Fig. 1

(A) Western blot for Foxa2 in live tissues at indicated time points after birth. (B) *Hnf4a*-specific m⁶A-RIP-qPCR for Control and *Mettl3* icKO mouse liver tissues at 2 weeks after Tamoxifen injection. Primers for *Hnf4a* mRNA m⁶A positive and negative regions were the same as in Fig. 5E. (C) Heatmap for m⁶A “readers” in neonate and adult livers. Data were analyzed from RNA-sequencing results in Supplementary Table 3. (D) RT-qPCR analysis of *Igf2bp1* mRNA in liver tissues from mice of indicated age.

Major issues:

1. Fig. 1B: *Mettl3* is decreased significantly at day 1 even though the albumin promoter are not expressed until 1-2 weeks postnatally. Explain this discrepancy.

Response: Thank the reviewer for your kind suggestions. We checked published literature and found that Cre under the control of *Albumin (Alb)*-enhancer/promoter was expressed in close correlation with the expression pattern of endogenous *Alb* gene in hepatocytes, and *Alb* begins to express before birth (*Genesis*. 2000 Feb;26(2):149-50; *Genesis*. 2009 Dec;47(12):789-92). An early study by *C Postic et. al* demonstrated that targeted DNA recombination efficiency in *Alb-Cre* mouse liver was about 40% immediately after birth and gradually increased with age (*Genesis*. 2000 Feb;26(2):149-50). These studies demonstrated that *Alb-Cre* transgene expresses in perinatal hepatocytes and is a useful tool for studying postnatal hepatocyte development.

Our work demonstrated significant downregulation of *Mettl3* in cKO mice compared to control in postnatal day 1 livers (**Fig. 1b**). Then we tested Cre expression and observed over 20,000 times higher expression of *Cre* mRNA in *Alb-Cre* mouse livers compared to Control at P0 (new **Supplementary Fig. 1i**). In addition, the *Alb-Cre* mice we used were created by site-specific insertion of “*Alb-enhancer/promoter-Cre*” cassette into “safe harbor” locus H11, and thus there is less possibility to be silenced and more likely to maintain high expression.

2. Figure 1C: What is the postnatal day that these samples were collected?

Response: We apologize for the careless errors we made. Samples in **Fig. 1c** were liver tissues collected from 2-week-old mice. We have added these information in the Figure legend (please see **Page 32 line 678**) and carefully double-checked the entire manuscript accordingly.

3. Figure 1C shows only a very faint band in the *Mettl3* cKO while SFig. 1e shows an incomplete knockout of *Mettl3* in liver. What is the explanation for this difference? Hepatocytes should be purified and analyzed to establish that the *Mettl3* gene is

knocked out in hepatocytes and the extent of knockout.

Response: Special thanks for the reviewer's careful reading and kind suggestions. We analyzed the images of original **Supplementary Fig. 1e** and found that the loading control Gapdh in *Mettl3* cKO liver was more than that of control. We conducted western blot again and updated the figure (please see new **Supplementary Fig. 1g**), which also showed a very faint band as in **Fig. 1c**. We also purified mouse hepatocytes and analyzed the protein level of Mettl3 as the reviewer suggested. The results showed almost complete deletion of Mettl3 in mouse hepatocytes from cKO individuals (new **Supplementary Fig. 1e-1f**). Slight residual Mettl3 detected in cKO livers may be explained by intact Mettl3 expression in other cell types (immune cells, hepatic stellate cells, endothelial cells *et al.*) within the liver.

4. Page 10, line 210: "...most Hnf4 α target genes involved in liver development, such as Apoa2, Apoc3, Cyp8b1, and Mttp.....". There is no evidence that Apoa2, Apoc3, Cyp8b1, and Mttp are involved in liver development. Which are the HNF4A target genes that are responsible for the liver phenotype and lethality phenotype? This should be determined by rescue experiments.

Response: We apologize for the improper expression in the original manuscript and thank the reviewer for pointing this out. We have now amended this in the manuscript (please see **Page 10 line 218 to Page 11 line 220**). What we want to say is that "RNA-sequencing data showed that along with downregulation of *Hnf4a*, most Hnf4a target genes, such as *Apoa2*, *Apoc3*, *Cyp8b1*, and *Mttp*, were repressed in *Mettl3* cKO individuals".

Previous studies showed that Hnf4a is essential for both hepatocyte specification in fetal liver and liver maturation and function in postnatal development (*Genes Dev.* 2000 Feb 15;14(4):464-74; *Mol Cell Biol.* 2001 Feb;21(4):1393-403; *Nat Genet.* 2003 Jul;34(3):292-6). Most aspects of hepatocyte function were affected by Hnf4a deletion, including epithelial formation, hepatic glycogen storage, bile acid homeostasis, and lipid metabolism (*Mol Cell Biol.* 2001 Feb;21(4):1393-403; *Nat Genet.* 2003 Jul;34(3):292-6; *Proc Natl Acad Sci U S A.* 2006 May

30;103(22):8419-24). Meanwhile, many Hnf4a target genes, such as Apoa2, Apoc3, Cyp8b1, Mttp, Cyp7a1, Ntcp, and several genes responsible for cell junction and adhesion, were significantly reduced upon Hnf4a deletion. Moreover, disrupting some of these Hnf4a targets led to liver damage at specific aspects. For example, deletion of Apoa2 in mouse liver resulted in Hypotriglyceridemia (*J Biol Chem.* 1994 Sep 23;269(38):23610-6), and hepatic ablation of Mttp led to moderate hepatic steatosis (*J Clin Invest.* 1999 May;103(9):1287-98). However, none of these studies phenocopies all aspects of *Hnf4a* knockout mice, especially the lethality. Moreover, no one validates the function of these targets with rescue experiments on Hnf4a-deficient mice. Therefore, we think it will be pretty meaningful to clarify the Hnf4a target genes mainly responsible for the liver injury and lethality phenotype with rescue experiments on Hnf4a-null mice in the future.

To further strengthen our conclusion that Hnf4a is the primary mediator of *Mettl3* function in liver development, we conducted rescue experiments using AAV8 to overexpress Hnf4a under the control of a liver-specific promoter (thyroxine-binding globulin, TBG) (AAV8-TBG-Hnf4a) on *Mettl3* cKO mice (Fig. 6j). First, we injected AAV8-TBG-Hnf4a through superficial temporal vein at day two after birth and collected liver tissues at two weeks. We found that overexpressing Hnf4a alleviated liver damage caused by hepatic *Mettl3* knockout compared to AAV8-Ctrl, evidenced by increase in numbers of proliferating hepatocytes and reduced hepatic steatosis after Hnf4a overexpression (please see new **Fig. 6k-6m** and **Supplementary Fig. 7m**). However, we did not see long-term benefits on mortality. This may attribute to the rapid dilution of AAV caused by the vigorous hepatocyte division within four weeks after birth (*Hepatology.* 1995 Sep;22(3):906-14). Then we overexpressed Hnf4a by AAV8-TBG-Hnf4a through tail vein injection at four-week-old *Mettl3* cKO mice. The results showed that Hnf4a overexpression significantly prolonged the life span of *Mettl3* cKO mice (please see new **Fig. 6n**). These results further demonstrated that Hnf4a is the primary factor mediating the function of *Mettl3* in liver development. We have added these data into the new **Fig. 6j-6n** and modified the text accordingly (please see **Page 13 line 267-282**).

5. As the authors note, the liver phenotype and the lethality are much more severe than that found with the hepatocyte-specific *Hnf4a*-null mice, also made with Alb-Cre. This indicates that there are other factors involved that are modified by *Mettl3* that have a role in liver development. RNA-seq should be carried out on hepatocyte-specific *Hnf4a*-null mice and *Mettl3* cKO mice.

Response: Many thanks for the review's kind suggestions. We fully agree with the reviewer that other factors modified by *Mettl3* besides *Hnf4a* also involved in liver development. Now we compared the transcriptome signatures of *Mettl3*-deficient livers and *Hnf4a*-deficient livers with our RNA-seq data of *Mettl3* cKO mouse livers and previously published RNA-seq data generated from hepatocyte-specific *Hnf4a*-null mice (*Genes Dev.* 2020 Aug 1;34(15-16):1039-1050). As expected, a large proportion of genes showed overlapped expression patterns between *Mettl3* cKO and *Hnf4a*-null liver tissues (**Attached Fig. 2A**). GSEA analysis also showed similar expression profile changes in *Mettl3* cKO and *Hnf4a*-null conditions, evidenced by concomitant enrichment of multiple transcription factor terms (**Attached Fig. 2B**). Also, targets of *Hnf4a* are among the significantly repressed pathways in both groups (**Attached Fig. 2B**). Notably, both up-regulated and down-regulated gene numbers are higher in *Mettl3* cKO livers than *Hnf4a*-null livers, indicating more potent effects caused by *Mettl3* knockout. Furthermore, although many liver development-related genes showed comparable downregulation in both *Mettl3* cKO and *Hnf4a*-null samples, some showed more obvious changes in *Mettl3* cKO livers, such as *Hpn*, *Rhbdd3*, and *Stat5b* (**Attached Fig. 2C**).

Attached Fig. 2

(A) Diagrams showing the overlapped changes of gene expression between *Mettl3* cKO and *Hnf4a*-null mouse liver tissues. (B) GSEA analysis of RNA-sequencing data showing significantly enriched shared in *Mettl3* cKO livers and *Hnf4a*-null livers. (C) Histogram showing representative genes down-regulated in both *Mettl3* cKO and

Hnf4a-null mouse liver (5 genes on the left) or *Mettl3* cKO mouse livers only (3 genes on the right).

6. The evidence (Figure 6H) that m⁶A modification of mRNA increases its degradation is not convincing. The decrease in mRNA is modest and could be the result of actinomycin-induced liver toxicity. An RNA-seq could be carried out to see if the effect is non-specific. It is just as likely that another mRNA or a miRNA that is modified by *Mettl3* alters levels of *Hnf4a* mRNA. Or the possibility cannot be ruled out that the modification alters *Hnf4a* pre-RNA processing, transport of the *Hnf4a* mRNA to the cytoplasm of *Hnf4a* mRNA translation.

Response: Many thanks for the review's kind suggestions. We conducted more experiments to strengthen our conclusion that m⁶A regulates *Hnf4a* levels mainly by controlling *Hnf4a* mRNA stability. We also analyzed alternative splicing and the nucleus-cytoplasm transportation of *Hnf4a* mRNA to exclude other possibilities.

Item 1: The reviewer pointed out that the mRNA degradation experiments were not convincing and concerned about the interference of actinomycin D-induced liver toxicity. Firstly, we checked published research and found that the trademark and concentration of actinomycin D in our work were the most commonly used in RNA decay experiments in different cell contexts, including hepatocytes (*Nat Cell Biol.* 2018 Mar;20(3):285-295; *J Hepatol.* 2021 Dec;75(6):1420-1433). To further rule out the possibility that drug toxicity induced a non-specific decrease of *Hnf4a* mRNA stability, we detected mRNA stability of other genes, such as *Hlx*, *Dbp*, and *Foxa2*, *et al.* in the same system using RT-qPCR (**Attached Fig. 3A-3C**). Most transcripts showed a rapid and significant decrease after actinomycin D treatment. Moreover, as the reviewer suggested, we conducted RNA-sequencing to analyze the global changes of mRNA half-life in hepatocytes from control and *Mettl3* cKO mice. Consistent with previous studies (*Nature.* 2014 Jan 2;505(7481):117-20; *Science.* 2015 Feb 27;347(6225):1002-6), knockout of *Mettl3* enhanced mRNA stability globally, especially for m⁶A-modified genes (please see new **Supplementary Fig. 7g, h**). We then analyzed the half-life time of a total of 131 genes in the liver development

pathway catalog (downloaded from Gene Ontology Resource, GO: 0001889) and found that only *Hnf4a* and another 10 genes showed decreased mRNA half-life when *Mettl3* was knocked out. Most genes (including *Cited2*, *Cebpa*, *Notch2*, and *Dbp* et al.) were increased or unchanged (please see new **Supplementary Fig. 7i**). We also tested the mRNA stability of *Hnf4a* in HepG2 cells with METTL3 knockdown and observed similar results as that in mouse primary hepatocytes (please see **Supplementary Fig. 7e, 7f** and new **Fig. 6f, 6g**). Then we checked previous publications and found that some genes also showed a moderate decrease when treated with actinomycin D in other contexts (*Oncotarget*. 2015 Mar 10;6(7):5041-58; *EMBO J*. 2020 Oct 15;39(20):e104514), which may be caused by longer half-life of these genes. Along with authentic m⁶A deposition and Igf2bp1 binding on *Hnf4a* mRNA, we concluded that the higher decay rate of *Hnf4a* mRNA was specifically caused by *Mettl3* knockout in developing mouse livers. We have added the corresponding data into the new **Fig. 6f, 6g**, and **Supplementary Fig. 7e-i** and modified the text accordingly (please see **Page 11 line 229 to Page 13 line 265**).

Attached Fig. 3

(A-C) RT-qPCR analysis of *Hlx* (A), *Dbp* (B), and *Foxa2* (C) mRNA levels in primary hepatocytes treated with actinomycin D for indicated time.

Item 2: The reviewer was concerned about the detailed mechanism m⁶A controlling *Hnf4a* mRNA levels, which was also concerned by Reviewer 1. We have conducted further experiments and demonstrated a clear mechanism that reduced binding of m⁶A “reader” IGF2BP1 is responsible for decreased *Hnf4a* mRNA stability in *Mettl3* cKO livers (please see new **Fig. 6h, 6i**, and **Supplementary Fig. 7j**), thus giving evidence

that m⁶A on *Hnf4a* mRNA can directly modulate the mRNA stability. In addition, alternative RNA splicing showed no differences in RNA-sequencing data from Control and *Mettl3* cKO liver (data not shown), and distribution of *Hnf4a* mRNA in nuclear and cytoplasm was also not affected by *Mettl3* knockout (new **Supplementary Fig. 7b-d**). Only mRNA stability showed significant changes in primary hepatocytes and HepG2 cells upon *Mettl3* deletion (new **Fig. 6f** and **6g**). Therefore, though we could not exclude the co-existence of other possibilities, such as another mRNA or miRNA contributing to *Hnf4a* mRNA stability as the reviewer mentioned, we think IGF2BP1-mediated stabilization is one of the major mechanisms *Mettl3* maintaining *Hnf4a* level in hepatocytes. Please also see **Answer to Item 3 by Reviewer 1 (Page X line X)** for detailed results and descriptions.

7. Fig. S6A, Supplementary Table 3: The RNA-seq of the Control and *Mettl3* cKO livers should be compared with hepatocyte-specific *Hnf4a*-null livers.

Response: Many thanks for the review's kind suggestions. We have now conducted these analyses as the reviewer suggested and observed that a large proportion of genes showed overlapped expression pattern between *Mettl3* cKO and *Hnf4a*-null liver tissues. Please see **Answer to Item 5 by the same Reviewer (Page 13 to Page 15)** for detailed results and descriptions.

8. There is a large variability in the HNF4A protein in Figure 6D. These blots should be quantified.

Response: We thank the reviewer for pointing this out. As the reviewer suggested, we have now conducted quantitation and added the data into the new **Fig. 6e**.

9. Figure 7: *Hnf4a* mRNA and its target gene mRNAs should be measured in this model.

Response: Many thanks for the review's kind suggestions. We analyzed the mRNA level of *Hnf4a* and its target genes of control and *Mettl3* icKO mouse livers by RT-qPCR and included the results in the new **Supplementary Fig. 8e-k**. Both *Hnf4a*

and its target genes were unchanged in *Mettl3* icKO livers compared to Control at different time points.

10. m⁶A modification levels should be measured in the adult *Mettl3*^{fllox/fllox}/*Alb-Cre*^{ERT2} (*Mettl3* icKO) mice with and without tamoxifen.

Response: Many thanks for the review's kind suggestions. We purified mRNA from Control (*Mettl3*^{fllox/fllox}/*Alb-Cre*^{ERT2} without tamoxifen) and *Mettl3* icKO (*Mettl3*^{fllox/fllox}/*Alb-Cre*^{ERT2} with tamoxifen) mouse livers and conducted LC-MS/MS to quantify the m⁶A levels. The results showed significant downregulation of bulk m⁶A level in *Mettl3* icKO mice liver (please see new **Fig. 7f**).

11. Does knockout of *Mettl3* in adult mice alter liver regeneration or response to chemically-induced liver injury?

Response: Many thanks for the review's kind suggestions. We are also curious whether knockout of *Mettl3* in adult mice alters liver injury and repair, though it minimally affects adult liver homeostasis. We found that *Mettl3* icKO mice (knockout of *Mettl3* in adult mice) showed more severe damage than Control mice in the CCl₄-induced liver injury model, evidenced by higher liver injury indicators (AST and ALT) and more TUNEL⁺ apoptotic cells (**Attached Fig. 4A-4C**). However, icKO mice also showed more Ki67⁺ proliferating hepatocytes 48 hours after 70% hepatectomy, suggesting better regenerative ability than control (**Attached Fig. 4D-4F**). We are still delineating the dynamics and detailed mechanism of this discrepancy and hope this work will give us a more comprehensive understanding of the role of *Mettl3* in the liver.

Attached Fig. 4

(A-C) Mice were intraperitoneally injected with 7.5 μ l/g 20% CCl₄ diluted in olive oil, and the livers were collected 24 hours later. (A) Serum levels of alanine aminotransferase (ALT) and aspartate aminotransferase (AST) (n = 4 for each group). (B) Representative photographs of HE and TUNEL staining of Control and *Mettl3 icKO* liver sections. Scar bar, 100 μ m. (C) Quantification of TUNEL⁺ cells in (A). (D-F) Mice received 70% hepatectomy, and samples were collected 48 hours later. (D) Serum levels of ALT and AST (n = 4 for each group). (E) Representative photographs of HE staining and Ki67 IHC staining of liver sections. Scar bar, 100 μ m. (F) Quantification of Ki67⁺ cells in (D).

12. Page 14, line 299: “In fact, most of the abnormalities we found in *Mettl3 cKO* mice (including lipid deposition in liver, increased bile acids in serum, liver injury and lethality in young adults) phenocopied that of hepatic *Hnf4 α* knockout mice.” Provide a citation for this statement.

Response: We thank the reviewer for carefully reading our manuscript and for your

kind suggestions. New citations (*Mol Cell Biol.* 2001 Feb;21(4):1393-403) have been added to the manuscript. Please see **Page 17 line 360 Ref #28**.

Minor issues:

13. Nomenclature: There are no symbols used in gene, mRNA, and protein nomenclature. Hnf4 α , Hnf1 α , Ppara α , and Cebpa α should be Hnf4a, Hnf1a, Ppara, and Cebpa.

Response: We appreciate the reviewer for pointing out these errors. We apologize for the careless mistakes we made in the original manuscript. All nomenclature has been revised now.

14. Figure 7F: The numbers in this table should be reduced to three significant figures.

Response: Many thanks for the review's kind suggestions. We have now revised the original **Fig. 7F** according to the reviewer's suggestion. We chose three indexes (AST, ALP, and Cholesterol) to put in the new **Fig. 7h** and the rest in the new **Supplementary Fig. S8d**.

15. METTL3 regulates heterochromatin in mouse embryonic stem cells (PMID: 33505026). This paper should be discussed as a possible mechanism that gives rise to the phenotype in the hepatocyte-specific *Mettl3*-null mice.

Response: We thank the reviewer for kindly reminding us of the newly published interesting research (*Nature.* 2021 Mar;591(7849):317-321). We have revised the discussion section and added the new citation (please see **Page 17 Line 365-372**).

16. It is unconventional and distracting for figures to be re-cited in the Discussion, other than a model figure.

Response: We are grateful for the reviewer pointing out our lack of standardization. We have now revised the discussion section and removed the re-citation of figures in the discussion section. Again, we thank the reviewer for pointing this out.

17. Figure 8 could be deleted or put in Supplemental figures.

Response: Many thanks for the review's kind suggestions. We have now put the graphic abstract into the new **Supplementary Fig. 9**.

REVIEWERS' COMMENTS

Reviewer #2 (Remarks to the Author):

I have no further comments

Reviewer #3 (Remarks to the Author):

The authors have addressed most of the reviewer concerns, and provide additional evidences to support their conclusion that m6A promote Hnf4a mRNA stability. However, the authors may notice that the most majority of the literatures support the principles in m6A field that m6A promote target mRNA degradation.

At least, the authors may need to tone down their conclusion that m6A may also regulate liver development via other targets/pathways other than Hnf4a. The authors may also could discuss to reconcile the discrepancy regarding m6A mechanism on how target mRNA stability was promoted rather than conventionally decreased by m6A.

REVIEWERS' COMMENTS

Reviewer #2 (Remarks to the Author):

I have no further comments.

Response: We appreciate the Reviewer's positive comments on our efforts. Many thanks for the Reviewer's careful evaluation of our work.

Reviewer #3 (Remarks to the Author):

The authors have addressed most of the reviewer concerns, and provide additional evidences to support their conclusion that m⁶A promote Hnf4a mRNA stability. However, the authors may notice that the most majority of the literatures support the principles in m⁶A field that m⁶A promote target mRNA degradation.

At least, the authors may need to tone down their conclusion that m⁶A may also regulate liver development via other targets/pathways other than Hnf4a. The authors may also could discuss to reconcile the discrepancy regarding m⁶A mechanism on how target mRNA stability was promoted rather than conventionally decreased by m⁶A.

Response: We thank the Reviewer for evaluating our work carefully and giving us valuable suggestions. We agree with the Reviewer that Hnf4a may not be the only target mediating the role of m⁶A in regulating postnatal liver development. Other pathways may also be controlled by m⁶A and contribute to this process.

As the Reviewer mentioned, m⁶A modification was conventionally considered to decrease mRNA stability for most genes in a YTH family-recognized pattern (*J Mol Biol.* 1978 Sep 25;124(3):487-99; *Nature.* 2014 Jan 2;505(7481):117-20; *Cell Stem Cell.* 2014 Dec 4;15(6):707-19; *Science.* 2015 Feb 27;347(6225):1002-6). However, recent studies found that a portion of genes showed reduced RNA stability with m⁶A removal, and the transcripts of these genes might be read by a distinct family of m⁶A readers, the insulin-like growth factor 2 mRNA-binding proteins (IGF2BPs; including IGF2BP1/2/3). Thus, it seems that "reader" types recognizing m⁶A modification determine the fate of mRNA in a specific context.

We performed RNA-seq for WT and *Mettl3* cKO primary hepatocytes treated with

actinomycin D for different periods. The results also showed that only a few transcripts showed shorter half-life in cKO cells, while most genes became more stable compared to WT hepatocytes (Supplementary Figure 7g-7i), consistent with previous reports (*Nature*. 2014 Jan 2;505(7481):117-20, *Nat Cell Biol*. 2018 Mar;20(3):285-295). Although *Hnf4a* showed decreased mRNA stability in cKO cells, some differentially expressed genes involved in liver development, like *Foxm1*, were more stable when *Mettl3* was knockout (Supplementary Table 5). *Foxm1* level was upregulated in liver diseases like liver cirrhosis and HCC, and overexpression of *Foxm1* was reported to lead to spontaneous liver injury (*J Hepatol*. 2012 Sep;57(3):600-12., *Genes Dev*. 2004 Apr 1;18(7):830-50, *Cell Mol Gastroenterol Hepatol*. 2020; 9(3): 425–446). We also observed an increased *Foxm1* mRNA expression in *Mettl3* cKO livers collected at 4 weeks (Supplementary Table 3). Thus, decreased m⁶A in *Mettl3* cKO mice livers may promote *Foxm1* mRNA stability by a YTH-recognized approach, lead to increased *Foxm1* level, and probably also contribute to defects in postnatal liver development. Therefore, m⁶A may also regulate the liver development process through an mRNA degradation-promoting pattern by other targets, as the Reviewer mentioned.

Besides RNA stability, *Mettl3* knockout may also lead to defects in liver development by regulating transportation, translation, and splicing of mRNA and biogenesis of miRNA. Therefore, we modified our manuscript accordingly, toned down the conclusion, and clarified that *Hnf4a* might not be the sole target of m⁶A in liver development (*please see Page 17, line 365-368*).